# A new ape from Türkiye and the radiation of late Miocene hominines

Ayla Sevim-Erol [1✉], David R. Begun [2✉], Alper Yavuz[3], Erhan Tarhan[4], Çilem Sönmez Sözer [1], Serdar Mayda [5], Lars W. van den Hoek Ostende[6], Robert M. G. Martin [2] & M. Cihat Alçiçek[7]

Fossil apes from the eastern Mediterranean are central to the debate on African ape and human (hominine) origins. Current research places them either as hominines, as hominins (humans and our fossil relatives) or as stem hominids, no more closely related to hominines than to pongines (orangutans and their fossil relatives). Here we show, based on our analysis of a newly identified genus, *Anadoluvius*, from the 8.7 Ma site of Çorakyerler in central Anatolia, that Mediterranean fossil apes are diverse, and are part of the first known radiation of early members of the hominines. The members of this radiation are currently only identified in Europe and Anatolia; generally accepted hominins are only found in Africa from the late Miocene until the Pleistocene. Hominines may have originated in Eurasia during the late Miocene, or they may have dispersed into Eurasia from an unknown African ancestor. The diversity of hominines in Eurasia suggests an in situ origin but does not exclude a dispersal hypothesis.

[1] Ankara University, Faculty of Languages History and Geography, Department of Anthropology, Ankara, Türkiye. [2] Department of Anthropology, University of Toronto, Toronto, ON, Canada. [3] Mehmet Akif Ersoy University of Science and Letters, Department of Anthropology, Burdur, Türkiye. [4] Hitit University Faculty of Science and Letters, Department of Anthropology, Çorum, Türkiye. [5] Ege University Fakulty of Science, Department of Biology, İzmir, Türkiye. [6] Naturalis Biodiversity Center, Leiden, The Netherlands. [7] Pamukkale University, Department of Geology, 20070 Denizli, Türkiye. ✉email: aylasevimerol@gmail.com; david.begun@utoronto.ca

The origin of the hominines is among the most hotly debated topics in paleoanthropology. The traditional view, ever since Darwin, holds that hominines and hominins originate in Africa, where the earliest hominins are found and where all extant non-human hominines live. More recently a European origin has been proposed, based on the phylogenetic analysis of late Miocene apes from Europe and Central Anatolia[1–3] The fossils described here attest to a lengthy history of hominines in Europe, with multiple taxa in the eastern Mediterranean known for at least 2.3 Ma[4–7] Our phylogenetic analysis, based on the new specimens described here and a large sample of other fossil and extant hominoids (Supplementary Note 1, Tables 1, 2), supports previous research confirming the hominine status of the eastern Mediterranean apes[2,3,8–20] Our most parsimonious phylogenetic results suggest that hominines in the eastern Mediterranean evolved from dryopithecins in central and western Europe, though there are alternative interpretations[21–24]. Either way, the oldest known hominines are European. They may have dispersed into Europe from ancestors in Africa, only to become extinct[22] However, the more likely and more parsimonious interpretation is that hominines evolved over a lengthy period in Europe and dispersed into Africa before 7 Ma.

For some time, the only known late Miocene ape from Anatolia was *Ankarapithecus*, which is alternatively described as a stem hominid or a pongine[25–27], but not a hominine. It is easily distinguished from *Ouranopithecus* and *Graecopithecus* from Greece and Bulgaria[25–27] In 2007, a new species of *Ouranopithecus* was described from Çorakyerler in central Anatolia[28]. Since then, thousands of vertebrate fossils have been recovered at Çorakyerler, including a well-preserved ape partial cranium[29] (Fig. 1) The *O. turkae* holotype, a fragmented palate, was originally distinguished from *O. macedoniensis* in its shorter premaxilla, narrower palate, morphologically similar (homomorphic) upper premolars (as opposed to P3 being more triangular than P4), smaller male canines and possibly larger size[28]. However, recovery of the new cranium and our reanalysis of the published material requires a reassessment of this conclusion and justifies the naming of a new genus of Miocene hominine. *Anadoluvius* is distinguished from other eastern Mediterranean apes in the palate, face, neurocranium, mandible, dental root and root canal configuration, and in dental crown proportions and morphology. *Ouranopithecus*, *Anadoluvius* and *Graecopithecus* may be members of an evolving lineage, with the new data from Çorakyerler further supporting the hominine affinities of these taxa. Hominines were more diverse in the late Miocene of the eastern Mediterranean than previously understood, with a known range from at least 9.6–7.2 million years ago. This conclusion, along with the recent reassessment of the Pyrgos Vassilissis and Azmaka specimens reveals a hitherto unappreciated diversity of late Miocene apes in the eastern Mediterranean.

## Results and discussion

**Systematic paleontology.** Order Primates Linnaeus, 1758
 Infraorder Catarrhini Geoffroy, 1812
 Superfamily Hominoidea Gray, 1825
 Family Hominidae Gray, 1825
 Subfamily Homininae Gray 1825
 *Anadoluvius* gen. nov.

*Synonymy*. *Ouranopithecus* Bonis and Melentis: Güleç et al. [28]

*Type species*. *Anadoluvius turkae* comb. nov. Sevim Erol et al. 2023.

*Etymology*. Anadolu is the modern Turkish word for Anatolia and Anatolian.

*Holotype*. CO-205, a fragmented but largely complete male palate with LI$^1$-M$^3$ and RC-M$^2$ (Supplementary Figs. 1, 2).

*Paratypes*. CO-300 (RM$_2$); CO-305 (male mandibular fragment with RC-M$_1$); CO-710 (female mandibular fragment with RP$_3$-M$_2$); CO-2100 (RI$^1$); CO-2800 (female partial cranium with RC-M$^2$, portions of the right maxilla, maxillary frontal processes, frontal maxillary processes and most of the frontal bone) (Fig. 1; Supplementary Figs. 3–5) Detailed specimen descriptions and a revised diagnosis for this new taxon appear in the Supplementary Notes 2, 3. The hypodigm is curated in the Department of Anthropology, Ankara University. All samples used in this analysis are listed in Supplementary Tables 1, 2. Measurements are provided in Supplementary Tables 3, 4. Supplementary Note 4 presents the results of a comprehensive quantitative analysis of

**Table 2 Hominine and pongine synapomorphies (18 taxa, ordered).**

| Hominine | Pongine |
| --- | --- |
| Supraorbital torus | Supraorbital margin |
| Frontal sinus/glabella | Interorbital space |
| Ethmoidal sinus | Lateral orbital pillar surface |
| I$^1$ marginal ridge shape | Nasal bone length |
| C implantation | Nasal bone breath |
| Upper premolar length | Clivus length |
| Lower molar buccal cristids | Clivus orientation |
| Supraorbital torus | C inclination |
| | Incisive canal |
| | Incisive foramen size |
| | Upper premolar crests |
| | P$_3$ hypoproto-postprotocristid |

**Table 1 Hominine and pongine synapomorphies (18 taxa, unordered).**

| Hominine | Pongine |
| --- | --- |
| Supraorbital torus | Supraorbital margin |
| Frontal sinus/glabella | Lateral orbital pillar surface |
| Ethmoidal sinus | Nasal bone length |
| I$^1$ marginal ridge shape | Nasal bone breath |
| C implantation | Clivus orientation |
| Upper premolar length | C inclination |
| Lower molar buccal cristids | Upper premolar crests |
| | P$_3$ hypoproto-postprotocristid |

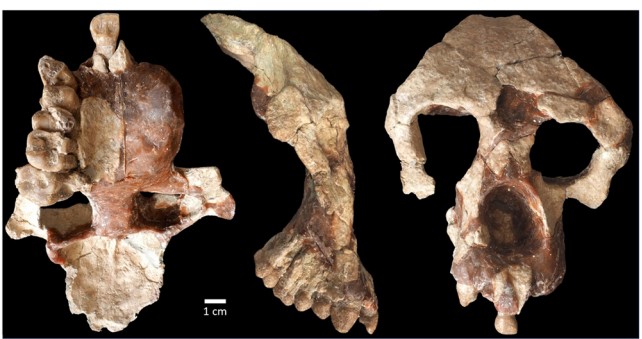

**Fig. 1 CO 2100/2800.** A female partial cranium. From left to right, palatal, right lateral and anterior views.

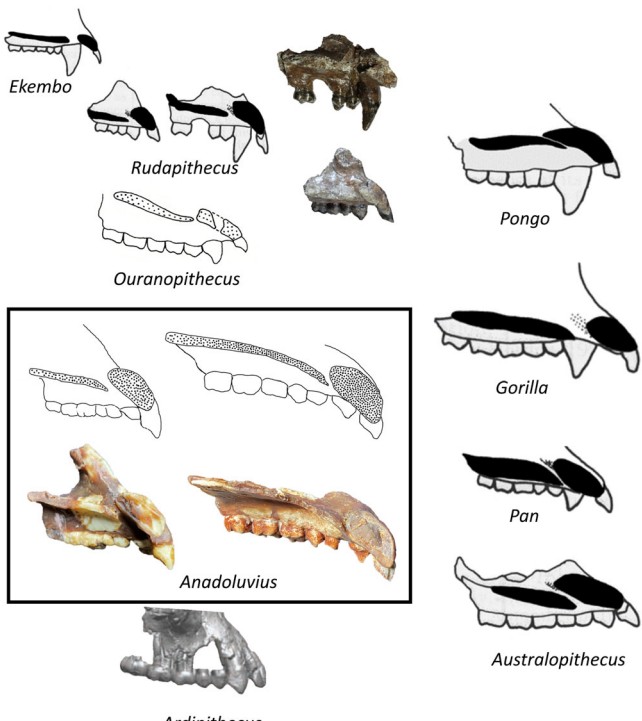

**Fig. 2 Cross sectional anatomy of the palate in *Anadoluvius* and other hominids (not to scale).** *Ekembo* and extant hominids redrawn from[31] *Rudapithecus* modified from[40]. *Ouranopithecus* redrawn from[41] based on a ct scan. The *Ardipithecus* specimen, modified from[42], is a surface rendering derived from ct scans and does not show the cross section but the lateral aspect. The *Ekembo* specimen is based on BMNH 16664, the holotype of *Ekembo nyanzae*. The *Rudapithecus* specimens are RUD 12, a female, and RUD 44, a male. The photographs to the right of the line drawings of *Rudapithecus* are the original specimens. The *Anadoluvius* specimens are CO-2100/2800 (female, left) and CO-205 (male, right), with photographs of casts of the reconstructed specimens (see SI for details of the reconstruction.) Line drawings of *Anadoluvius* are original to this work.

the *Anadoluvius* hypodigm. Supplementary Note 5 provides details of the phylogenetic analysis. Supplementary Note 6 provides historical, geological and biochronologic background.

*Cranium.* The CO-2100/2800 partial cranium (Fig. 1, Supplementary Figs. 2, 3) was recovered with some crushing and displacement of several broken pieces (see SM for restoration details and detailed description.) The frontal bone is nearly intact, missing only portions within the temporal fossa and the squama approaching bregma. This distinguishes it from the most complete facial specimen of *Ouranopithecus* (XIR-1), which is broken just beyond the superior orbital margins and preserves almost nothing of the frontal squama (Supplementary Fig. 2). Though damaged, the position and orientation of the premaxilla is better preserved in CO-2100/2800 than in CO-205 and *Ouranopithecus* (XIR-1 and RPl 128), confirming previous interpretations of a stepped and overlapping morphology in these specimens[1–3,11,13–15] (Fig. 2).

The premaxilla of *Anadoluvius* is short and vertical compared with *Pan*, *Pongo*, and australopithecines, and is most like *Gorilla* and dryopithecins, being relatively short in the alveolar portion but expanded nasally to overlap with the palatine process of the maxilla (Fig. 2) The incisor alveoli are positioned along the mesial transverse plane of the canine crowns (Figs. 1, 2; Supplementary Fig. 5) In two specimens of *Ouranopithecus* (RPl 128 and XIR 1) the upper incisors are well anterior to the canines (Supplementary Fig. 5) In NKT 89 the premaxilla is severely damaged, but the

posterior edge of the lateral incisor appears to be aligned with the anterior transverse plane of the canines, a position most like *Anadoluvius*. The frontal bone of *Anadoluvius* differs strongly from that of *Ouranopithecus* in the smooth biconvex squama of the former, contrasting with a broad concavity above glabella in the latter. The superior orbital margins of *Ouranopithecus* are broad, rounded and slightly projecting while they are sharp and flat in *Anadoluvius*.

*Mandible.* A principal components analysis based on mandibular measurements available for *Graecopithecus*, *Ouranopithecus* and *Anadoluvius* is presented in Supplementary Fig. 6 and Supplementary Data 1). The Çorakyerler, Nikiti 1 and *Graecopithecus* mandibles are separated from each other, especially along PC 2, and from *Ouranopithecus*, illustrating the diversity present in these samples. Supplementary Data 1 includes the data matrix, summary statistics, scores, and loadings.

*Anadoluvius*, like *Graecopithecus*, and NKT 21, has a relatively narrow mandible compared with the combined sex sample of *Ouranopithecus* (Supplementary Fig. 7a). Supplementary Fig. 7b compares relative mandibular corpus breadth at each tooth position ($P_3$-$M_2$) in *Anadoluvius*, *Graecopithecus*, and *Ouranopithecus*. *Anadoluvius* is similar in mandibular robusticity at the premolar level but at the level of the molars it matches or strongly exceeds the maximum value in the other taxa (Supplementary Fig. 7b).

There is diversity in mandibular robusticity (breadth relative to height) and in dental size ratios as well among the samples of eastern Mediterranean apes (Supplementary Fig. 8a–e). *Anadoluvius* is distinct from *Graecopithecus* in all mandibular and dental ratios. *Anadoluvius* falls beyond the range of variation of *Ouranopithecus* in relative corpus breadth at $M_1$-$M_2$ (Supplementary Fig. 8a) and $M_2$ size (Supplementary Fig. 8d). Interestingly, NKT 21 falls outside the *Ouranopithecus* range in relative $P_4$ length and $M_2$ size (Supplementary Fig. 8d–f). It has a relatively short symphyseal-molar distance, at the 25% quartile for the Ravin sample, and a relatively robust mandible at $M_1$-$M_2$, at the 75% quartile for *Ouranopithecus* (Supplementary Fig. 8a, c). In *Graecopithecus* the symphysis is positioned closest to the molars, just barely in the range of the Ravin sample. In summary, in most quantitative comparisons *Anadoluvius* is distinguished from *Ouranopithecus* and *Graecopithecus*.

*Tooth roots and enamel thickness.* Figure 3 shows the root and root canal morphology of CO-300, the male mandible of *Anadoluvius* (see Methods for segmentation details). Unlike *Ouranopithecus*, the distal roots of $P_3$ to $M_1$ in *Anadoluvius* and *Graecopithecus*[11] are single fused roots with two root canals (Fig. 3 and Supplementary Table 5).

Like *Ouranopithecus*, *Anadoluvius* has thick enamel. Supplementary Fig. 9 illustrates ranges of variation in relative enamel thickness (RET) in the $M_2$ of Miocene, Plio-Pleistocene and living hominoids. *Anadoluvius* has thicker enamel than most Miocene apes, falling at the upper end of the range in *Afropithecus*[30] Its RET is greater than RPl 641, an $M_3$ of *Ouranopithecus*. The relationship between $M_2$ and $M_3$ RET is variable in hominoids[30], but their ranges of variation always overlap. *Anadoluvius* falls well above the ranges in extant hominids and within the *A. afarensis* and *A. africanus* 75% quartiles.

*Canine size.* The results of an ANOVA examining lower canine relative size is presented in Supplementary Table 7. In canine size relative to the geometric mean *Ouranopithecus* is significantly different from extant African apes in having relatively small canines. The relative size of the CO-305 mandibular canine (0.53) is equal to the mean of *Ouranopithecus* males and at the low end of the range of variation in *Pan* males and females.

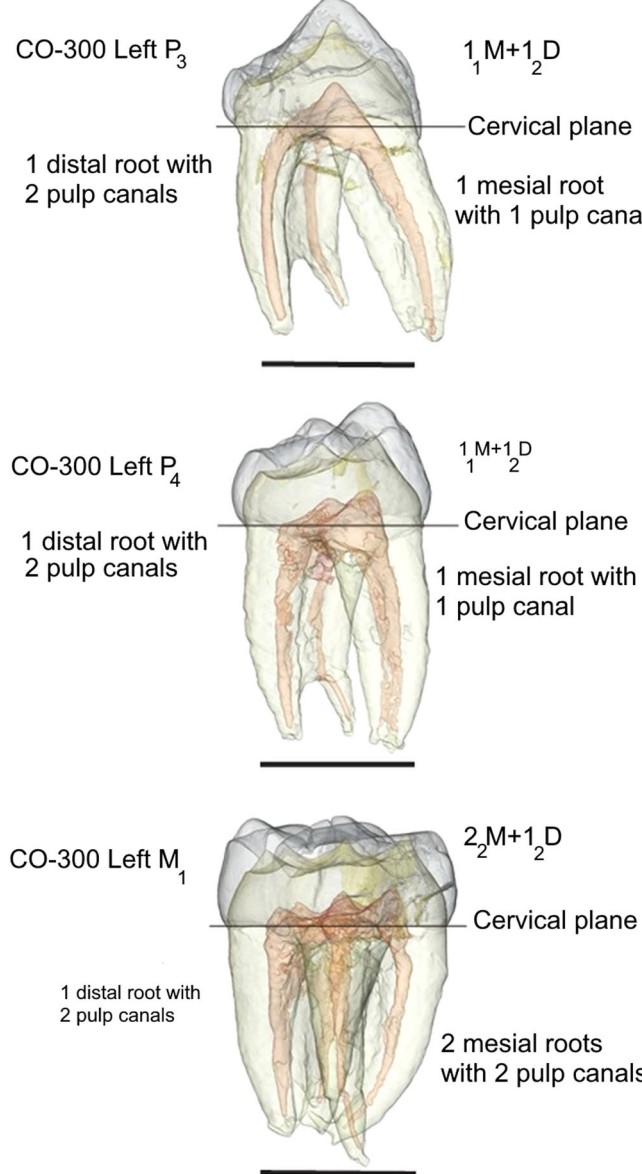

**Fig. 3 3-D reconstruction of the left P3 to M1 of CO 300, showing the root, root canal and pulp chamber configurations.** Supplementary Table 5 for a comparison of root formulae. Scale =10 mm.

Canines are small in the Balkan/Anatolian sample compared with other fossil and extant apes including *Ardipithecus*, being more consistent with the ranges in *Australopithecus*. The ratio of lower canine to $M_2$ size (canine maximum ln x bd/$M^2$ maximum ln x bd) in *Anadoluvius* is lower than in any male and most females except *Australopithecus* (Supplementary Fig. 10a). The Balkan/Anatolian specimens have relatively small canines compared with African apes when scaled to the individual geometric means (Supplementary Fig. 10b and Supplementary Tables 6, 7; geometric mean of 12 variables). Scaled $M_2$ size is large and beyond the range of variation of African apes in *Graecopithecus*, *Anadoluvius*, and Nikiti 1while Ravin de la Pluie *Ouranopithecus* is intermediate between the other fossil apes and *Gorilla* (Supplementary Fig. 10c). Canine size is compared to each tooth position in Supplementary Fig. 11a, b.

The results of a cladistic analysis using a data matrix of 112 characters and between 18 and 23 taxa are presented in Fig. 4, Tables 1, 2 and in the supplementary materials (Supplementary Figs. 12–14; Supplementary Data 2, 3; Supplementary Note 5). All four cladograms are strict consensus. The analysis was run both with all characters unordered and 21 of the 112 characters ordered (see Methods and Supplementary Note 5 for character matrix assumptions). Both analyses yield similar results. The tree topologies are identical, and predictably the tree values are lower in the analyses using ordered characters. All but the cladograms that include the taxa with more than 80% missing data recover a clade that includes Eurasian apes and hominines to the exclusion of pongines and stem hominids (Fig. 4). The potential problem of using data from published sources as opposed to direct observation is illustrated in the results for *Sahelanthropus*. This taxon, universally attributed to the Homininae and most commonly to the Hominini, is never recovered in these positions in these analyses, being consistently a stem hominid. *Sahelanthropus* could be coded for 72% of the characters. The potential for coding inconsistent with the criteria used to code other taxa is therefore larger than for *Orrorin*, which is also coded from the literature, but for which only 29% of the characters could be coded.

We mapped synapomorphies and Bremer support values onto two consensus cladograms (Supplementary Figs. 12, 13). Table 1 lists the hominine synapomorphies of the cladogram (ordered and unordered) with the fewest missing data (18 OTUs). A phylogeny consistent with a large majority of the cladograms presented here appears in Fig. 5.

Supplementary Fig. 14 shows the results of the analyses of all four taxon sets with all character states ordered. Unlike the unordered and partly ordered analyses, the cladograms with differing OTUs vary widely. The 19 OTU (including *Sahelanthropus*) fully ordered analysis is consistent with previous ones while the 18 and 20 OTU analyses result in a pongine clade including all European and Anatolian taxa, which contrasts with all previous cladistic results from analyses with large data sets and numerical cladistic methods (e.g. TNT, PAUP, etc). The 20 OTU analysis includes several other unconventional results such as an *Orrorin-Anadoluvius* clade that, along with *Ouranopithecus* as the sister to pongines, and a pongine clade that includes all European and Anatolian taxa. The 23 OTU analysis remains highly unresolved. Given the diversity of results of these all ordered analyses and the broader consistency of the analyses in which the data matrices were either unordered or partly ordered, we consider the latter to be more reliable (see Methods for a discussion of our rationale on ordering character states in this analysis).

The consensus cladogram with the three taxa represented by <20% of the character matrix (23 OTUs) is uninformative, given the low level of resolution. The cladograms resulting from the unordered or partly ordered analyses, which exclude these taxa (*Chororapithecus*, *Samburupithecus* and *Graecopithecus*) all recover a clade that includes *Anadoluvius* and *Ouranopithecus* as sister taxa, which in turn is either the sister taxon to the dryopithecins or in an unresolved polychotomy with the dryopithecins and the crown hominines. *Nakalipithecus*, which has been interpreted as a potential ancestor of *Ouranopithecus*, is outside the crown hominids in this analysis, as suggested elsewhere[1,31] *Ankarapithecus* is identified as a pongine and the widely accepted *Pan*-hominin clade is supported as well.

*Ouranopithecus*, *Graecopithecus* and *Anadoluvius* share a suite of derived characters of the jaws and dentition that support their status as a distinct clade. Although *Graecopithecus* could not be included in the analyses that yielded well resolved phylogenies, due to its many missing data (90%), all previous analyses of *Graecopithecus* associate it phylogenetically with *Ouranopithecus*[1–3,5,11,13,15,32]. The core attributes of the Balkan/ Anatolian late Miocene apes are large, thickly enameled molars,

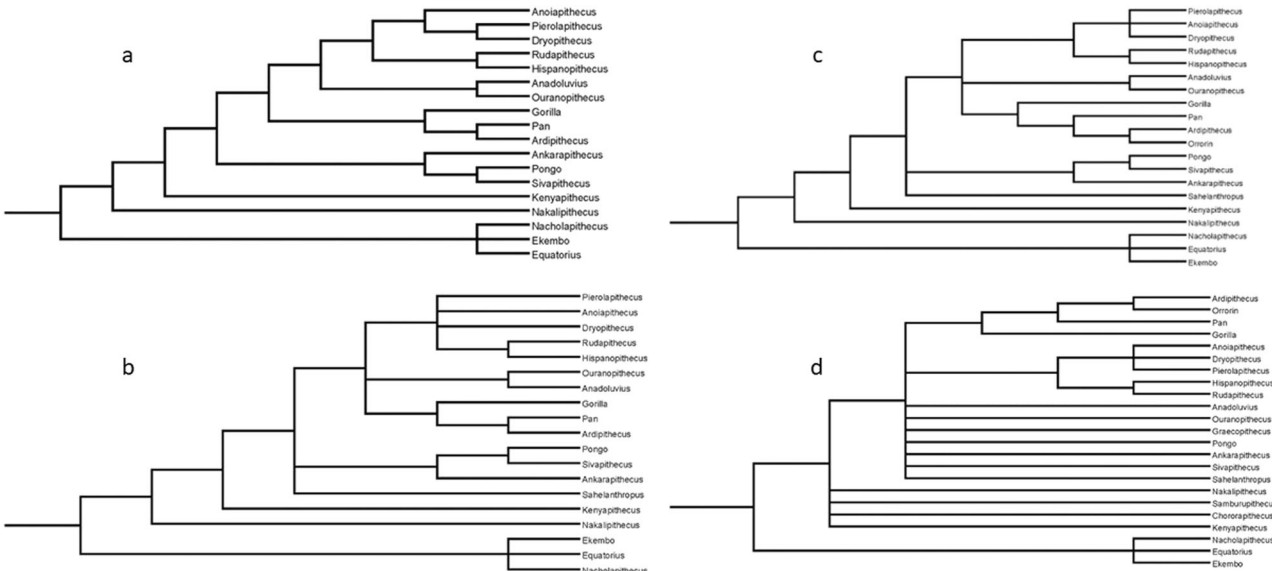

**Fig. 4 Strict consensus cladograms.** The four taxon sets each produced cladograms with the same topology whether character states were left unordered or a subset were ordered (see Methods and Supplementary Note 5 for details). **a** 18 OTUs. The four taxa with the fewest codable character states (*Graecopithecus*, 10%, *Chororapithecus*,13%, *Samburupithecus*, 18%, and *Orrorin*, 29%) were excluded, as was *Sahelanthropus*. Both *Orrorin* and *Sahelanthropus* were coded from published descriptions, which introduces uncertainty (DRB, who coded all characters in this analysis, was unable to code characters from these taxa through direct observation). **b** 19 OTUs, with *Sahelanthropus* added. **c** 20 OTUs with *Orrorin*. There is a decrease in resolution with the inclusion of *Sahelanthropus* and *Orrorin* but the tree topologies are otherwise consistent. *Sahelantthropus* is always recovered as a stem hominid and *Orrorin* as a hominin. The first three cladograms all recover a hominine clade that includes the thickly enameled Balkan taxa and the dryopithecins. **d** 23 OTUs, including all taxa. Little resolution remains among hominids, with recognized clades (pongines) unresolved. This cladogram also fails to recover *Ouranopithecus* as a hominine, which is otherwise a common result in previous analyses. Bremer support values, character states, character definitions and the character matrix (nexus) are all included in Supplementary Note 5 and Supplementary Data 3.

transversely robust mandibles, small canines, and large size. However, among these taxa there is diversity. *Graecopithecus* is distinguished from *Ouranopithecus* (*contra* 35) in its relatively large $M_2$ compared with both the $M_1$ and corpus breadth, its more vertical mandibular symphysis and in details of root morphology[1,11,15] *Anadoluvius* has the same lower dental root formula ($P_3$ to $M_1$) as *Graecopithecus* and both differ from *Ouranopithecus*. The frontal bone between the superior orbital margin and the anterior temporal line is preserved in XIR-1 (*Ouranopithecus*), which is sufficient to show that it was more vertically oriented than in CO-2100/2800. *Anadoluvius* is further distinguished from *Ouranopithecus* and other non-hominin hominines in having mesiodistally shorter canines (as was probably the case in *Graecopithecus* based on canine root size and shape) that lack mesial grooves and lingual cingula (unknown for *Graecopithecus*).

In quantitative attributes, the PCA (Supplementary Fig. 6) illustrates the overall distinctiveness of *Anadoluvius* compared with Balkan apes. The male *Anadoluvius* mandible CO-300/305 is distinguished from male *Ouranopithecus* in many metric comparisons (canine-$M_2$ ratio, canine/geometric mean, relative mandibular breadth, symphyseal-molar distance, relative $P_4$ length and size, and relative $P_4$, $M_1$ and $M_2$ size (Supplementary Fig. 8a–f). *Anadoluvius* is distinguished from male *Ouranopithecus* in canine size relative to postcanine tooth size at every tooth position for both upper and lower tooth rows (Supplementary Fig. 11). *Anadoluvius* is distinguished from *Graecopithecus* in relative mandibular breadth, symphyseal-molar distance, $P_4$ length, $M_2$ relative size and $M_2$ size relative to corpus breadth (Supplementary Fig. 8). In addition, the mandibular arch is wider and relative mandibular breadth larger at every dental position in *Anadoluvius* (Supplementary Fig. 7). Unfortunately, *Nakalipithecus* is insufficiently preserved to be included in most of these

quantitative analyses, which are scaled using the geometric mean. *Nakalipithecus* lacks one or more of the quantitative attributes needed to generate the geometric mean used in this analysis. We ran an analysis of $M_2$ size relative to mandibular corpus breadth at the level of mid $M_1$, which are data that have been published for *Nakalipithecus*. In this ratio the *Nakalipithecus* specimen, which is probably male, falls within the range of gorilla females, *Pan* of both sexes, and female *Ouranopithecus*, outside the range of gorilla males, *Ouranopithecus* males, Nikiti, *Graecopithecus*, and *Anadoluvius* (Supplementary Fig. 8f).

*Ouranopithecus* and *Anadoluvius* lack shared derived characters of the pongines (greatly elongated premaxilla substantially or completely overlapping the maxillary palatine process, expanded zygoma, tall orbits, narrow interorbital space, reduced or absent ethmoidal frontal sinus, circumorbital costae.) There is no evidence for their inclusion in Ponginae.

*Ouranopithecus* and *Anadoluvius* share with dryopithecins (European middle and late Miocene apes with affinities to *Dryopithecus*) a series of characters found among hominines[1–3] These include a ventrally rotated palate, a stepped subnasal fossa, broad, flat nasal aperture base, short nasal bones, nasal aperture apex superior to the infraorbital margins, robust lateral orbital pillars, frontal sinus expanded below nasion, incipient supraorbital torus, more horizontal frontal squama[1–3] The phylogenetic significance of some of these shared attributes is disputed, particularly concerning the dryopithecins[9,24]. However, there is broad agreement that *Ouranopithecus* shares enough derived characters with hominines to warrant inclusion in that taxon[1–3,8–20] (Table 1).

The phylogenetic results presented here regarding *Ouranopithecus* and its sister, *Anadoluvius*, are consistent with many previous analyses[1–3,8–20] They are robust in terms of the number of synapomorphies and Bremer support values for many clades

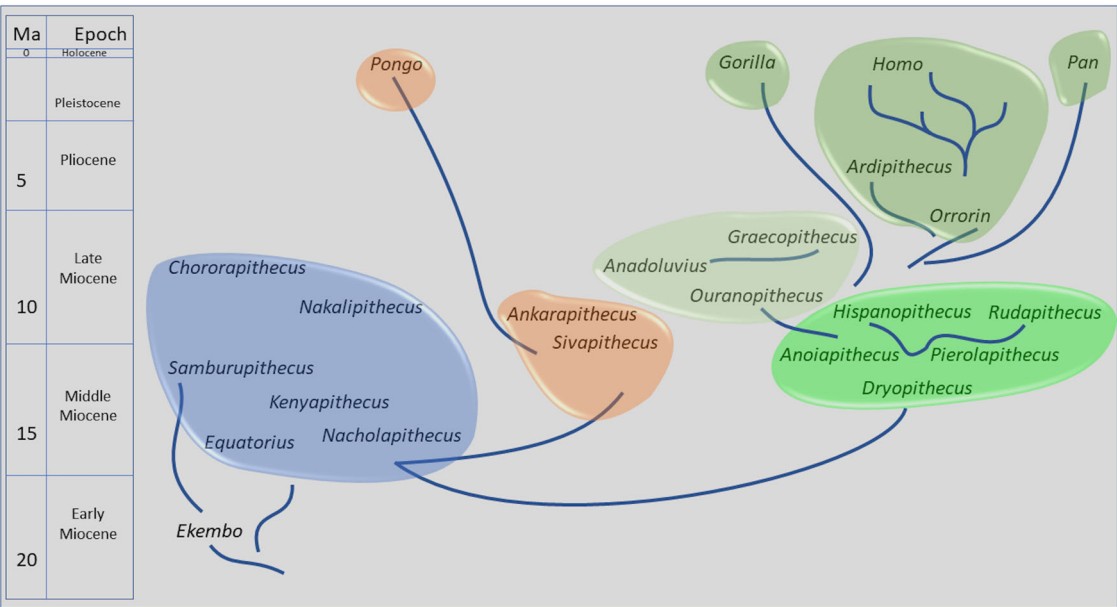

**Fig. 5 A phylogeny of the taxa included in this analysis consistent with most of the cladograms presented here.** Taxa are positioned in chronological order without regard to geography, with most taxa only known from a limited time span. Exceptions are *Ekembo* and *Sivapithecus*, with longer time ranges, which are positioned roughly when they are most abundant, in both cases about mid-way in their known time ranges. The different colored "puddles" represent hominid clades and/or stages of evolution. These can also be imagined as pools of related species in somewhat delimited space and time with broad ancestor-descendant relationships. The lines are disconnected to reflect the difficulty in identifying actual ancestor-descendant relationships, but that these relations can be estimated between "puddles". Blue puddle taxa are stem hominids and are all confined to Africa. Among these taxa the relations of *Samburupithecus* and *Chororapithecus* are unresolved in the cladograms except in so far as they are excluded from the clade that includes all Eurasian taxa and crown hominins. Other lines of evidence suggest that these taxa are members of the early or middle Miocene radiation of early apes (see text). The orange puddles are the pongines, which probably have their origin within the middle Miocene puddle, although not necessarily any of the taxa included here (another taxon, *Griphopithecus*, known from Europe and Türkiye, would be a member of the blue puddle but was not included in the cladistic analyses). While both are pongines, *Ankarapithecus* lacks derived features shared by *Sivapithecus* and *Pongo*, so the line representing the relationship between the latter two bi-passes *Ankarapithecus*. The three green-shade puddles represent the hominine clades as defined here. Bright green are the dryopithecins, with the younger taxa *Rudapithecus* and *Hispanopithecus* depicted as closely related and descendant from any of the older taxa or an unknown taxon sharing attributes with these three. The light green puddle includes the Balkan and Anatolian taxa, likely to have descended from somewhere in the dryopithecin puddle. Possible ancestor-descendant relationships are depicted in this puddle. The darker green puddle represents the crown hominines. The various lineages diverge from unknown ancestors, but probably a member of either of the older green shade puddles. Gorillas diverge first, followed by chimpanzees and humans. *Orrorin* and *Ardipithecus* are depicted in a manner consistent with their sister clade status, without implying a direct ancestor-descendant relationship.

(Supplementary Figs. 12, 13). The results also recover the widely accepted relations among crown hominids and hominines and relations within all fossil clades. The Bremer support values for the hominid clade as defined here (15-16) are extremely strong. The hominine clade as defined here is moderate to strong (2-3), and very strong (4 to 8) for the pongine clade (Supplementary Figs. 12, 13). European and eastern Mediterranean apes are classified as hominine in three of four consensus cladograms. It fails to be resolved only in the analysis of all 23 taxa, including *Samburupithecus*, *Chororapithecus* and *Graecopithecus*, missing 82%, 87% and 90% of the data respectively. The latter cladogram also fails to distinguish between pongines and hominines.

The relations among the dryopithecins are consistent with most detailed analyses focused on this group, as are the relations within the Asian clade (*Ankarapithecus*, *Sivapithecus* and *Pongo*) and the crown hominines[1–3,8–20]. The consistencies of these cladograms in many details with previous research lend credibility to these results. Recent analyses by (9 and 10) are broadly similar in their results. Among the most parsimonious cladograms reported in ref. [10] at least one also recovers a hominine clade that includes European fossil taxa[9]. also recovers *Ouranopithecus* as a hominine though in that analysis the dryopithecins are recovered as stem hominids.

Other taxa have been linked with *Ouranopithecus* or hominines. Three fossil apes from Africa, *Nakalipithecus*,

*Samburupithecus* and *Chororapithecus*, broadly overlap in time with *Ouranopithecus*, *Graecopithecus* and *Anadoluvius*[21–23] *Nakalipithecus* has been identified as potentially ancestral to *Ouranopithecus*[22]. However, *Nakalipithecus* differs in many details of dental morphology from *Anadoluvius* and *Ouranopithecus* (see Supplementary Note 3). As noted, *Nakalipithecus*, represented in the data matrix by 43% of the total number of characters, consistently falls outside the crown hominids, which fails to support the hypothesis of an ancestral-descendant relationship with *Ouranopitheus*. It has been hypothesized that both *Chororapithecus* and *Ouranopithecus* have phylogenetic affinities with gorillas[8,22,23,33]. However, *Chororapithecus* is readily distinguished from both *Anadoluvius* and *Ouranopithecus* (see differential diagnosis), and its affinity with gorillas has been questioned[1,34] A close phylogenetic relationship with gorillas is not supported by our results[21]. claim that *Samburupithecus* has phylogenetic affinities with African apes and humans, though this conclusion has also been challenged by[1,35], who conclude that *Samburupithecus* is not a hominid but instead a vestige of the early Miocene proconsuloid radiation. As with *Chororapithecus* and *Nakalipithecus*, *Samburupithecus* falls outside the crown hominid clade.

*Diversity and paleobiogeography.* A comprehensive review of the taxonomy and phylogeny of late Miocene apes is needed, given

recent discoveries and reinterpretations. Here we focus on the diversity and paleobiogeographic implications of *Anadoluvius* (Supplementary Fig. 15; Supplementary Table 8). *Anadoluvius* and *Ouranopithecus* share attributes with other European middle and late Miocene hominids that distinguish them from late Miocene ape fossils from Africa and the broadly contemporaneous pongines *Sivapithecus*, *Ankarapithecus*, and *Khoratpithecus* (Supplementary Note 3). *Ouranopithecus* from Ravin de la Pluie and Xirochori are dated to 9.6 and 9.3 Ma respectively[6,7]. *Ouranopithecus* from Nikiti 1 is dated to 8.9 Ma[6,7]. Çorakyerler is dated to 8.7 Ma (Supplementary Figs. 16, 17; Supplementary Table 9 and Supplementary Note 6). The Nikiti mandible and maxilla are distinct from *Ouranopithecus* from the more northern Macedonian sites and may represent a different taxon[13]. This possibility, which needs further study, is interesting in terms of regional evolution as Nikiti 1 is likely to be slightly older than Çorakyerler[6,7,29]. Nikiti, *Anadoluvius* and *Graecopithecus* are distinguished from *Ouranopithecus* in our PCA and in having a greater degree of canine reduction, elongated $P_4$ and large molars relative to mandibular corpus size. *Anadoluvius* and *Graecopithecus* are distinguished from *Ouranopithecus* in root morphology (unknown in NKT 21.) NKT 21 and *Graecopithecus* have inferior transverse tori positioned posterior to the mesial edge of the $M_1$ (unknown in *Anadoluvius*), while the torus is anterior to the $M_1$ in *Ouranopithecus*. *Graecopithecus*, which is considerably younger (~7.2 Ma)[4], has been shown to differ from *Ouranopithecus* in morphology that replicates differences between late Miocene apes and early hominins, such as reduced relative canine size and premolar root morphology[11] However, the Nikiti specimens have not previously been included in these comparisons.

Quantitative and qualitative comparisons reveal many differences among Balkan/Anatolian taxa, indicative of a greater diversity of late Miocene eastern Mediterranean hominids than previously recognized. Hominines appear to have been present and diverse for millions of years in the late Miocene of Europe. Based on the large number of qualitative and quantitative differences among the samples from Macedonia, Attica and Anatolia, we conclude that they represent at least three hominine genera, *Ouranopithecus*, *Graecopithecus*, and *Anadoluvius*. The diversity of hominines in the eastern Mediterranean mirrors that among australopithecines in the Plio-Pleistocene hominin record in Africa. In the phylogenetic analysis presented here the Balkan/Anatolian taxa are in the sister clade of crown hominines. The fact that dryopithecins are also classified as hominines in this analysis suggests that there was an in situ evolution of thickly enameled late Miocene eastern Mediterranean hominines from more thinly enameled precursors in central and western Europe, though this conclusion has been challenged[9,24]. *Pierolapithecus*, *Anoiapithecus*, *Dryopithecus*, *Hispanopithecus* and *Rudapithecus* all share attributes with extant hominines and are distinguished from pongines such as *Sivapithecus* and *Ankarapithecus* (Tables 1, 2)[1–3,36,37], (but see refs. [9,24] for alternative views). A clade that includes both thinly and thickly enameled taxa, in this case the dryopithecins and the Balkan/Anatolian apes, has a parallel in Africa with *Ardipithecus* and australopithecines and with *Pan* and hominins.

The parallel evolution of thickly and thinly enameled members of a clade in Africa and Europe is not proof that the late Miocene European apes are all hominines, but it does make this hypothesis, supported by the results of our cladistic analysis, plausible. It is possible that the generally more thinly enameled dryopithecins and the later occurring thickly enameled Balkan/Anatolian hominines do not share an ancestor-descendant relationship and represent separate dispersal events into Europe from Africa (e.g. ref. [22]), though this is less parsimonious

biogeographically and contrasts with the results of the phylogenetic analysis presented here. While independent dispersal events are possible, we regard the in situ European hypothesis as more likely and more parsimonious given the current evidence. Other independent lines of evidence are also consistent with the widespread presence of hominines in Europe[38].

## Conclusions

Eastern Mediterranean apes previously assigned to *Ouranopithecus* span at least 2.4 million years with a geographic range roughly equal to the area between *Hispanopithecus* in Spain and *Rudapithecus* in Hungary (Supplementary Fig. 6) In the eastern Mediterranean, as in Western/Central Europe, multiple genera are present. A preliminary analysis of stable isotopes from single tooth samples of *Ouranopithecus* from RPl and *Anadoluvius* and larger samples of hipparionins suggests cooler, drier conditions at Çorakyerler and a reliance on $C_3$ plants for both ape taxa[39]. Their thick enamel, robust jaws and, in the younger samples from Nikiti 1, Çorakyerler and Pyrgos Vassilissis, evidence of canine reduction, recalls the earliest African hominins. A clarification of the significance of the similarities between the Balkan/Anatolian samples and late Miocene African hominins will have to await further discoveries.

Eastern Mediterranean hominines may represent a terminal radiation arising from one or more older hominines in Europe, analogous to the radiation of *Paranthropus* presumably from an *Australopithecus*-like ancestor. Alternatively, given that European hominines most closely resemble gorillas[1–3,8,33,36,40], it is possible that this represents a radiation of early members of the gorilla clade, mirroring the middle and late Miocene radiation of pongines in Asia[37]. It is also possible that European hominines represent terminal lineages of successive dispersals of hominines from Africa[22], though there is no evidence of multiple lineages of hominines between 13 and 10 Ma in Africa, and this hypothesis is not supported by the results of the phylogenetic analysis presented here. Finally, some researchers have focused attention on differences among eastern Mediterranean apes, suggesting that multiple lineages are present, including the earliest known hominins[11,33] In this context it is worth noting that *Graecopithecus* was only included in the phylogenetic analysis with all taxa, which yielded a poorly resolved cladogram. *Graecopithecus* is grouped with hominids but not specifically with other eastern Mediterranean hominines. However, multiple analyses have concluded that *Graecopithecus* and *Ouranopithecus* are closely related, even if this has not been demonstrated with a formal cladistic analysis (see above and Supplementary Notes 3–6).

A comprehensive phylogenetic analysis is underway to test these competing hypotheses. Whatever the outcome, the sample of ape fossils from Çorakyerler demonstrates that great ape diversity in the eastern Mediterranean is greater than previously believed and that hominines had diversified into multiple taxa long before their first documented appearance in Africa.

## Methods

**Materials**. All fossils attributed to *Anadoluvius*, fossils of other taxa, and all geologic samples were recovered from the fossil locality of Çorakyerler (40°36'32"N, 33°38'01"E) in the Çankırı Basin of the central Anatolian Cenozoic basin complex, and are currently curated in the Department of Anthropology, Ankara University. Comparative samples are outlined in Supplementary Tables 1, 2. All fossils recovered at Çorakyerler since 1999 are permitted by the Culture and Tourism Ministry (Dr. Ayla Sevim Erol, PI, Re: E-94949537-160.01.01-3883433, Konu : Çankırı İli, Çorakyerler Fosil Lokalitesi Kazı İzni).

**Measurements**. Dental dimensions are standard mesiodistal and buccolingual measurements, except canine and $P_3$ measurements, which are crown maximum (major axis) and perpendicular dimensions. Mandibular breadth measurements were taken just below the margin of the alveolar process to allow for comparisons with CO-300, which lacks the corpus base. The distance between the symphysis

and the $M_1$-$M_2$ interproximal space is a non-standard measurement that allows for the comparison among all the specimens. It is the horizontal chord between the lingual surface of the symphysis at the alveolar edge between the central incisor alveoli and the $M_1$-$M_2$ interproximal space at the level of the alveolar margin. The geometric mean was calculated from the mandibular measurements and the three dental dimensions that could be measured for *Graecopithecus* (Supplementary Data 1). In cases in which mandibles were incomplete or crushed, mandibular arch breadth between antimeres was estimated by doubling the distance from the alveolar margin to the midline.

The diagnoses and descriptions here result from direct observation of original fossil material by ASE and DRB. All published specimens currently attributed to *Ouranopithecus* were examined, measured, and photographed. These observations were compared with direct observations of all published European middle and late Miocene hominids (numerous repositories, see ref. [36], see Supplementary Tables 1, 2 for explanation of the abbreviations), the type and palate of *Ankarapithecus* (MTA), a mandible (MTA) and a sample of isolated teeth (NHM) of *Griphopithecus* and *Kenyapithecus*, all published African Miocene catarrhines (NMK, NHM, ENM), large portions of the sample of *Sivapithecus* (Harvard, AMNH) and the IVPP sample of *Lufengpithecus* up to 1995 (Supplementary Table 2). Comparisons with Pliocene hominins are based on observations on original material of *Ardipithecus*, *Austalopithecus* and *Paranthropus* (NME, NMK) supplemented with high resolution casts of *Austalopithecus* and other Pliocene hominins. Data from original fossils were supplemented with high resolution casts and/or published descriptions and images of *Morotopithecus* and *Proconsul* from Uganda, *Orrorin*, *Sahelanthropus* and *Khoratpithecus*. Comparisons with extant hominoids are based on data collected from the institutions listed in Supplementary Table 1. All specimens were measured by DRB using digital calipers recorded to the nearest 10th of a millimeter (Supplementary Tables 3, 4).

CT images were processed by RMGM using well established methods. The complete image stacks of each tooth were filtered using a three-dimensional median filter with a kernel size of 1 followed by a mean of least variance filter with a kernel size of 1. The filtered image stacks were imported into Avizo 6.3 (www.thermofisher.com) where the enamel, dentine, pulp cavity, and cracks were segmented semi-automatically using the 3D voxel value histogram and grayscale values. In Avizo 6.3, the teeth were reconstructed from the segmentations as triangle-based surface models using the constrained smoothing parameter.

**Statistics and reproducibility**. Quantitative analysis was carried out using © PAST and © Excel for Microsoft 365. The PCA is based on 12 measurements of the mandible and dentition scaled by their geometric mean (Supplementary Data 1). The PCA matrix was set at variance-covariance. Eigenvalues and % variance, scores and loadings for each PC appear in Supplementary Data 1. All data needed to replicate our phylogenetic analysis, using our assumptions and constraints or any other constraints (weighting, ordering etc.), are provided in the supplementary data. The characters and their states are listed and defined in Supplementary Data 2, Supplementary Note 5 and the data matrix in TNT format appears in Supplementary Data 3.

**Phylogenetic analysis**. The matrix analyzed here is original to this research and was assembled by DRB based on direct observation of all available specimens of all taxa included in this analysis, except *Orrorin* and *Sahelanthropus*, which were scored from published descriptions. Scoring of *Ardipithecus* is based on observations of the original fossils supplemented with published descriptions. Since the focus of this analysis is late Miocene eastern Mediterranean hominids, we used Mesquite (Mesquite version 3.70 (build 940) Copyright (c) 1997-2021 W. Maddison and D. Maddison) to define multiple outgroup taxa (early Miocene *Ekembo* and middle Miocene *Equatorius* and *Nacholapithecus*). We performed multiple analyses, differing in included taxa and assumptions of character ordering. Characters were unweighted. We used a progression of OTUs based on representation in the data matrix and reliability of character scoring. From a total potential number of OTUs of 23, we progressed from 18 OUTs, excluding *Sahelanthropus* and *Orrorin*, which could only be coded from published descriptions, and *Samburupithecus*, *Chororapithecus* and *Graecopithecus*, with more than 80% missing data. The 19 OTUs include *Sahelanthropus* (72%), as noted, coded from published descriptions. *Orrorin* (29%) was added next (as with *Sahelanthropus*, coded from published descriptions), followed by all three poorly represented taxa. Each OTU set was analyzed using fully unordered, partly ordered, and fully ordered character states (see below for further details). The four topologies including 18, 19, 20 and 23 OTUs are reproduced in Fig. 4. As noted, the topologies are identical in both the partly ordered and unordered analyses. Results of the analyses using fully ordered character matrices on each OTU set appear in the supplementary data (Supplementary Fig. 14).

The partly ordered character matrices consist of roughly half the multistate characters (21 of 41) ordered. Wagner parsimony allows for reversals, so two state characters (71 of 112) were not ordered. Of the 20 multistate characters that were not ordered, all consisted of characters in which the outgroup (*Ekembo*, *Equatorius*, and *Nacholapithecus*) had the intermediate condition. These intermediate character states are polarized by definition as primitive. To order these characters would mean forcing the algorithm into a transformation series from intermediate to one extreme and then the other, which is both counter-intuitive from an evolutionary

perspective and less parsimonious. For example, ordering character 57, $I^2$ position relative to the nasal margin, requires a transformation series from the intermediate condition (in line with the nasal margin) to one extreme (e.g. mesial) and then the other (e.g. lateral). Allowing the algorithm to freely follow a transformation sequence from intermediate to one extreme or the other without having to pass through the opposite extreme seems more reasonable. Supplementary Note 5 lists all the characters, their states, definitions, and whether they are ordered in the partly ordered analyses.

Our decision to present the results of the analysis of unordered character matrices, which are always more parsimonious for these data, is based on our concern that ordering transformation series within a character limits the number of alternative character state transformations and presupposes a specified history of evolution for a particular character. It could be argued that because a transformation series results from genetic modification in ancestral-descendant populations, the ends of a transformation series must pass through intermediates, so that an ordering of character state transformations emerges inevitably from the evolutionary process. While it may seem intuitive that a character such as maxillary length should be ordered (going from short to intermediate to long), the fossil evidence indicates that this is not the case within this sample of taxa. Character state orders that seem "counter-intuitive" can be explained by the fragmentary and random nature of the fossil record. Over the course of hundreds of thousands or millions of years there is sufficient time for reversals or apparent "jumps" to occur without the preservation of intermediate steps in the fossil record. We know very little about the genetics of character state transformations. Intermediate states between thin and thick enamel are known because they are documented in different taxa, but this does not prove that all transformations for thick to thin in every lineage must have passed through the same intermediates. A short period of extended duration of amelogenesis from a thinly enameled ancestor may result in a final thickness that would be measured as thick. We simply do not know enough about the genetics of character formation to say with certainty that all character transformation series must pass through intermediates from one extreme to the other.

Other "counter-intuitive" character state transformation series include the development of the ectotympanic tube (absent, partial or complete), and lumbar vertebrae number (seven to six to five to four to three), none of which are supported by the majority of most parsimonious analyses of hominoids in numerous publications. Must these transformations have had intermediates that are simply missing from the fossil record or is it possible to go from, for example, seven to four to five lumbar vertebrae by simple conversion of vertebral type (e.g. thoracic to lumbar)? Humans have a median number of five lumbar vertebrae but four is not uncommon. Great apes have a median number of three but four is common. Given this variability a "jump" from three to five or vice versa is certainly feasible. We do not understand the genetics of these character states sufficiently to know how transformation occurs and if "jumps", which are more a function of how we define character states than anything else, are likely, probable or impossible. We need to remember that all character states are defined by the researcher and have no fundamental genetic basis. They are simply what we observe and deem to be informative. Most character states are amalgams of genetically determined attributes and processes controlled by one or more genes and their interactions. Finally, it could be argued that allowing a character state to "jump" to a subsequent state without an intermediate condition based purely on parsimony is akin to rejecting Darwinian gradualism in favor of Goldschmidtian "hopeful monsters". However, Goldschmidt was referring to species origins and not individual character states. Hypothesizing character state transformations that do not pass through intermediate states does not violate the basic principles of Darwinian evolution.

We choose not to weigh characters to avoid biasing the analysis by unduly emphasizing or de-emphasizing one character over another, since we do not see how weighing can be accomplished confidently in this character matrix.

All data needed to replicate this analysis, using our assumptions and constraints or any other constraints (weighting, ordering etc.), are provided in the supplementary data.

Consensus cladograms were recovered using TNT with the traditional search option and Wagner trees enabled [Willi Hennig Society; © Goloboff & Catalano (2006). Cladistics. DOI 10.1111/cla. 12160]. TNT was used to map synapomorphies and calculate Bremer support using the DOBREM script and calculate a strict consensus tree (Fig. 4 and supplementary data). The characters and their states are listed and defined in tables S2, 3 and the data matrix in TNT format appears in Supplementary Data 3.

**Reconstructions**. The holotype and all the paratypes were molded and cast by DRB using high resolution RTV silicone and polyurethane (Esprit Composite ™.) Molds were made of each of the separated portions of CO-2800 (premaxillary fragment, right maxilla, interorbital fragment, left orbital pillar and frontal.) CO-2100 (RI[1]) was found separately and is associated with CO-2800. Molds and casts were also made of this specimen. These casts were used to facilitate the reconstruction and will be made available to interested colleagues.

CO-205 (Supplementary Figs. 1, 5). The palate of *Anadoluvius* was recovered in two main fragments, two isolated teeth and some smaller fragments. The restoration of this specimen was relatively straight forward due to the preservation of multiple points along the midline. The right side preserves much of the palatine

process of the maxilla with two midpoints ($P^4$ and $M^2$- $M^3$), with the intermaxillary suture visible. A piece of the left alveolar process with parts of the alveoli of the $I^1$ and $I^2$ is also preserved, with the midline intact. Simple mirror-imaging allows for the unambiguous positioning of the alveolar processes, premaxillary fragment and incisors. The nasoalveolar clivus is not preserved beyond the portion of the palatal surface and the partial incisor alveoli. However, with the palatine process preserved to the midline it is possible to restore the entire premaxillary palatal surface with mirror imaging. Supplementary Fig. 1 illustrates issues with the reconstruction in ref. [28] corrected in our analysis.

CO-2800 (Fig. 1 and Supplementary Figs. 2, 3). The female partial cranium (CO-2800) was recovered intact in a block of relatively soft mudstone matrix (Supplementary Fig. 3) There was some distortion and slight displacement between portions separated by cracks. The matrix was carefully removed and replaced with plasticine modeling clay to ensure that the original positions of the displaced portions (incisor and premaxillary fragment, frontal nasal processes, left malar infraorbital surface and left orbital pillar) were maintained. Slight adjustments were made to bring conjoining surfaces into alignment though some plastic deformation remains.

The superior margin of the right orbit and the frontal bone in the region of the right temporal line are distorted. The frontal bone superior to the right orbit is pushed slightly superiorly and posteriorly and the right orbital pillar is tilted anteriorly. On the left side the specimen was fractured, which probably limited plastic distortion. A few millimeters separate the preserved portion of the nasal process of the maxilla from the maxillary process of the frontal bone on each side. They were positioned following the contours of the preserved surfaces, leaving little room for error. However, it is possible that the maxilla was slightly more ventrally rotated than in our final reconstruction. About 10 mm separate the premaxillary fragment from the palatine process of the maxilla. However, as it preserves portions of the alveoli for both upper central incisors and for the right $I^2$ this fragment could be accurately positioned in the midline, in alignment with other preserved midline structures (nasion and the intermaxillary suture). The nearly complete alveolus for the $RI^1$ is a perfect match for CO-2100. Casts of the separated portions of CO-2800 were used to produce a reconstruction with the tooth row, right nasal margin and palatine process, which are displaced but cannot be repaired on the original, repositioned and realigned (Supplementary Fig. 2) Some residual plastic distortion on the right side remains. For the final restoration of the original fossil the clay was replaced with casting wax, which is more resilient.

CO-305 (Supplementary Fig. 4). This mandible is damaged and cannot be restored reliably either on the original or with casts, given the extent of plastic deformation. However, it was possible to remove some matrix, exposing more of the canine, which is pushed into its alveolus. We were able to segment (Avizo) and virtually extract the canine from μct scan data (Supplementary Fig. 11a). Cleaning revealed that the alveolar portion of the symphysis is in its natural position relative to the right alveolar process of the corpus, allowing for comparisons with Balkan specimens. CO-300, an $M_2$ (Supplementary Fig. 4) was not included in the original hypodigm of *Ouranopithecus turkae*. It is a perfect match for CO-305 (wear, size and interproximal facet) and is described here together with the mandible.

**Reporting summary**. Further information on research design is available in the Nature Portfolio Reporting Summary linked to this article.

## Data availability

All data generated or analyzed during this study are included in this publication (and Supplementary Materials). All specimens from Çorakyerler are deposited in the Department of Anthropology, Ankara University. Accession numbers are provided in the column headings to Supplementary Tables 3, 4. Comparative samples are listed in Supplementary Tables 1, 2. The computed tomography scans are available from ASE or DRB on reasonable request. The new taxon has the following Life Science Identifier: urn:lsid:zoobank.org:act:FE6E46C7-BA39-428C-88E7-DC5FB2F5BE43.

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

## Acknowledgements

We are grateful to the reviewers and editors for their insightful comments that significantly improved this manuscript. DRB acknowledges with much gratitude the researchers, curators and collections managers who have provided access to fossils and comparative collections in their care. The authors thank Prof. Dr. Volkan Karabacak (Eskisehir Osman Gazi University) for the DEM images in Supplementary Fig. 14. The Çorakyerler excavations are supported by the Turkish Ministry of Culture and Tourism, the General Directorate of Cultural Heritage and Museums, Ankara University, and the Turkish Historical Society; we are thankful to all of them. DRB acknowledges support from NSERC (grant number RGPIN-2016-06761).

## Author contributions

A.S.E. is the PR of the project, secured funding for excavations and lab analysis and directed excavations. A.S.E. and D.R.B. are responsible for data curation, acquired funding and supervised research. D.R.B. conceptualized the analysis, conducted the investigation, developed the methodology, prepared the original draft and all subsequent versions. C.S.S., S.M., L.W.vdH. and C.A. contributed their analysis of geological and paleontological results. R.M.G.M. was responsible for the segmentation and analysis of the scans of the mandible. A.Y and E.T. participated in the excavations.

## Competing interests

The authors declare no competing interests.
