## [Peer Review File · Communications Biology]

Reviewers' comments:

Reviewer #1 (Remarks to the Author):

Thank you for the opportunity to review this manuscript. The authors present a new fossil Miocene ape cranium from Turkey, which they claim represents a new genus, suggesting "unappreciated diversity" of the late Miocene apes of the Mediterranean. Unfortunately, there were a number of major issues with this paper that render it unacceptable for publication and unprepared for a full review.

The original fossil should be shown with no reconstruction alongside the reconstruction to allow readers to independently evaluate its validity. As it is currently displayed, portions of the preserved fossil are "obscured by wax"

Phylogenetic analysis is insufficient and poorly explained

- "cladistic analysis using a data matrix of 112 characters selected to minimize missing data" Selected from what? Did you create a new character matrix or should this refer to a published matrix? If so, which one?
- What drove the choice of ingroup taxa? What about everything else between Ekembo and crown great apes?
- Which specimens comprised each of the fossil hypodigms? Did you code these fossils yourselves or were these character states pulled from another paper?

What were the 'mandibular measurements' that went into the PCA in Figure 5a? The text refers the reader to the methods section to find these measurements. I was only able to locate a mention of mandibular breadth, and a description for the "distance between the symphysis and the M1-M2 interproximal space". Surely the PCA was not conducted on 2 measurements?

Figures are not suitable for publication. Examples include:

- Figure 4: The colors are indistinguishable and the legend has inconsistent naming conventions (e.g. "Homo sapiens"& "Proconsult major" vs. "anamensis" & "boisei" vs. "Pan"& "Gorilla"). Same issues in figure 5b (with an "Ardi" thrown in).
- Figure 7: "see text" is not an acceptable figure caption.
- Figure S8 is just screenshots from a computer program (TNT).. That is not publishable.

Reviewer #2 (Remarks to the Author):

This is an interesting manuscript describing and naming a new hominoid ape genus from the material previously referred to as *Ouranopithecus turkae*. Overall, it is well-written, the descriptions are well-done, and I think that the authors are probably correct. The differences discussed and figured appear to be distinct enough (in my opinion) to warrant a new genus.

That being said however, the analyses provided are not very rigorous and need to be bolstered in order to "prove" their case. I have a couple major areas of concern that require substantial revisions before this paper should be published, and depending on what the revised analyses look like, they may or may not change my above assessment. I also have a number of additional more minor, but still significant comments and concerns as well. All of these are detailed in the attached annotated PDFs for the main text and Supplementary Information, and all of them need to be addressed, but I will briefly summarize them here:

Major issue 1: There are very few statistical tests in any of the comparisons, and Pan and Gorilla appear in some comparisons but not others. Because the authors are arguing that *Ouranopithecus* and *Anadoluvius* are hominines, the extant hominines need to be included in all comparisons so that we can properly judge the amount of variation that is typical within a hominine species and between genera for the features discussed here. There is no excuse for excluding the extant taxa in the comparisons, particularly Gorilla, which is more similar in size to the fossil taxa and has been noted to share similarities with them in the past. So these comparisons need to be added in before we can properly judge and have confidence in whether or not the differences presented here rise to the level of a separate genus rather than a separate species. Again, my sense is that they probably do, but the authors need to prove it with a broader taxonomic sample (i.e., Pan and Gorilla) for many of the comparisons.

Major issue 2: The phylogenetic analysis presented here is methodologically unsound and is thus not credible. There is no real justification for excluding taxa, and it sounds like (and I strongly suspect) a fishing expedition for a tree that the authors like. The citations provided to try and justify the exclusion of numerous relevant fossil taxa in fact do NOT support the argument the authors are making; to the contrary, the Pattinson et al. (2014) article in particular was a response to the severely flawed Darwinius analysis that made many of the same arguments about excluding incomplete taxa that the authors make again here! Instead, it has been demonstrated over and over again that missing data is not a serious problem in morphological systematics as long as character sampling is robust and some informative characters are sampled (Wiens, 1998; 2003a; 2003b; Fulton and Strobeck, 2006; de Queiroz and Gatesy, 2007; Wiens and Morrill, 2011; Pattinson et al., 2014). More broadly speaking, when has willfully excluding relevant data and information ever led to a better scientific hypothesis? Why would it be reasonable to exclude clearly relevant taxa because they are incomplete? The answer, of course, is that it is not reasonable. Furthermore, there are other serious problems in the cladistics methodology, taxon sampling aside. For example, leaving all characters unordered is a very poor way of modeling how evolution works through directional selection in most cases. Why aren't characters with clear evolutionary sequences ordered? If there is a population of apes that are small in body size, they would not suddenly evolve large body size without the population passing through an intermediate body size first through directional selection. That's how basic quantitative population genetics and directional selection works...body size, in this case, would be an ordered character with states of 0=small, 1=intermediate, and 2=large. And so it is only reasonable to model quantitative features such as nasal aperture breadth, nasal bone length, etc. in a similar way. Leaving them unordered is making an assumption about how evolution works that is almost certainly unreasonable. Again, for continuous traits, you don't just jump from one categorical state at the "small" end of the character transformation series to the largest categorical state in the transformation series without passing through the intermediate state, so why allow these types of things to go on in your matrix when running a phylogenetic analysis? Also, how were polymorphisms coded or handled? The authors don't ever say. And were any characters quantified and were any methods for coding continuous quantitative characters used? TNT allows the direct input of continuous data, or methods such as gap-weighted coding could be used to remove some of the subjectivity in a number of cases. I don't see any evidence that any of this has been done. How many Outgroups were assigned? Only Ekembo? Why? Numerous studies have documented that multiple outgroups help to improve accuracy...why only one here? It is unfortunate that Paleoanthropology continues to lag behind the rest of vertebrate paleontology in systematic methods, and the poor reasoning and methodology here seemingly ignores the past couple decades of research and reflects analyses more typical of the 1980s and early 1990s. The current analysis, its assumptions, and its character coding methods are obsolete and poorly justified; they should not be published here.

Fortunately, there are a couple of relatively easy solutions to this issue. One option is to revise the cladistic analysis to include all relevant taxa (or as many as possible), even if incomplete, add additional outgroups, and model characters more reasonably. That would be the bare minimum if the authors want to include a cladistic analysis here. The second viable option is to simply remove the analysis here and instead cite previous and recent analyses that suggest a similar position for

Ouranopithecus and/or Anadoluvius as hominines. For example the recent PhD thesis by Kelsey Pugh (Pugh, 2020) recovers a similar hominine position for Ouranopithecus (including *O. turkae* and *Graecopithecus*), or the Nengo et al. (2017) analysis (which is less ideal, because it doesn't include all of the relevant taxa discussed here) at least recovers trees consistent with what the authors are arguing in terms of Ouranopithecus being a hominine. There are probably others as well that I can't recall off the top of my head, but these trees and previous publications make the point the authors are trying to make and use much more modern cladistics methodology. I realize that some of these other analyses don't recover dryopithecines in the position that some of the authors favor, but again previous analyses that do can be cited if the authors want to push the biogeography argument and make the case that dryopithecines are also African great apes. Since *Nakalipithecus* isn't included in the analyses here, the biogeography argument is not a terribly strong one, but I think it's relevant and can be mentioned as a possibility that hominines evolved in Europe/Eurasia first. But it's also possible that they evolved in Africa (e.g., *Nakalipithecus*) and that *Ouranopithecus*, *Graecopithecus*, and *Anadoluvius* represent limited extensions out of Africa to the circum-Mediterranean rather than evidence of a migration back into Africa, and this possibility should be discussed as well. The overall point is that these arguments can be made by citing previous phylogenetic analyses rather than the subpar one being presented here.

More minor but significant comments:

Figure 1: The quality of the text in this image from GoogleEarth is poor. Please clean this up in Photoshop in the text make it more easily readable. You can leave the pins but remove the labels in GoogleEarth and add them in Photoshop instead. And the key in the upper right corner looks fuzzy and should be revised as well.

Figure 2 and all figures with specimens in them (both main text and SI): Please identify the sex of the specimens in the captions when known.

More broadly, rather than just showing *Anadoluvius* here, please provide a figure with a photo of the partial *O. macedoniensis* crania and CO 2100/2800 together somewhere (main text or SI) for direct comparison. And why not combine this Fig. 2 with Fig. S4 to show the original and new reconstructions side by side and with *O. macedoniensis*? This would make visual comparisons much easier for the reader.

Systematic Paleontology section: Likewise, when listing specimens, please indicate sex as male, female, or unknown whenever possible. E.g., CO-205, a fragmented but largely complete MALE palate with....CO-305 (MALE partial mandible with RC-M1), CO-2800 (partial FEMALE cranium) etc. etc.

Page 6, lines 115-122: This "short and vertical" premaxilla needs to be illustrated somewhere relative to the taxa you are mentioning so we can see this. I'm not sure I understand exactly what you are talking about here. It doesn't look very short and vertical in Fig. S4.

Also, the incisor alveoli don't really appear to be in line with the canines to me. They look anterior to me in the best preserved specimens or the reconstruction in Fig. 2 and S4. In any case, I generally agree that *Ouranopithecus* looks different from *Anadoluvius* here, but it would be nice to see this feature quantified. It should be possible to measure the distance between the canine to prosthion or something (in relative terms, of course). Also, please provide a photo of NKT 89 highlighting this feature in the SOM somewhere so the reader can see the similarity and judge for themselves.

Table 1: This notation is extremely difficult to read. Could this be presented in a clearer way? Maybe just present the number of mesial and distal roots in this paper and save the root canals for a separate table in the SI? Either that, or make better use of subscripts and superscripts like in Figure 3.

Page 10, PCA: How many measurements were included in the PCA? Which measurements? These need

to be provided or flagged somewhere so we know what's going into the PCA. From what I can gather, these are all RAW measurements, so PC1 is almost certainly just size (which is why it separates males and females). This is not very useful for separating taxa at the genus level here, in my opinion...or at least not useful just by itself. There should be a PCA using size-adjusted measurements as well so that we can see how the specimens differ with regard to SHAPE with size removed as much as is possible. And extant hominine taxa (Pan and Gorilla) should be included in the PCA as well so that we have some idea of the range of variation on the axes between living species and genera for comparison. Also, any discussion of the loadings should make sure to run a correlation of the first few PCs with the Geometric Mean of all measurements or some other correlate of overall size so that we know how size is loading on the various axes....even with absolute size being adjusted for, PC1 is often still correlated with size and therefore depicting allometry. In any case, this needs to be revised and a size-adjusted analysis needs to be included instead of or in addition to the current analysis.

Page 10, lines 172-178: Why is mandibular arch breadth the best denominator for comparison? Why not use a mandibular GMean as a size-adjustment instead? How do we know that Ouranopithecus doesn't have a relatively thick or thin mandible? And is canine size allometric? Gorilla males need to be included here for comparison since the fossils are closer to Gorilla than Pan in size. For m2 size and canine size, an ANOVA with post-hoc comparisons should be run between the Ravin males, Pan males, and Gorilla males to test for statistical significance here.

Figure 5: The colors for the various taxa/populations change in each panel. Please make the colors the same for each taxon throughout the figure for ease of comparison.

Figure 6 and discussion of Fig. 6 on page 12 lines 191-202: These are relatively small samples you are comparing, so it is difficult to draw conclusions without additional context. How do these fossil samples compare to the levels of variation seen within extant apes, particularly the extant hominines Pan and Gorilla? Pan and Gorilla need to be included for comparison in all of these box plots so that we can judge if the differences between ANadoluvius and Ouranopithecus exceed that seen within extant hominine species, particularly the similar-sized Gorilla. In addition, here again the color scheme in the Figure should be consistent throughout all the boxplots.

Pages 12-14: See my comments above for all the problems with the cladistic analysis. The citations provided don't support the removal of incomplete taxa and it is unclear whether some of the cited studies regarding coding issues for missing data in INVERTEBRATES are even relevant for vertebrates and mammals (e.g., the cited Sansom article). It also seems like at the bottom of page 12 that the authors are suggesting that most analyses that include as many taxa as possible are those that include many extant taxa and use morphological and molecular data. What are the citations for this statement? This is not really true for many analyses in primate and vertebrate evolution more broadly. For example, the Strait and Grine/Mongle et al. analyses of early hominins and extant great apes don't include any molecular data, include more fossil taxa than extant taxa, and include nearly all hominin taxa that are widely recognized, even if incomplete. More recent phylogenetic analyses in vertebrate and primate evolution certainly take advantage of both molecular and morphological data because numerous studies have demonstrated their effectiveness and higher degrees of accuracy, particularly when using a molecular backbone which partly eliminates the problem of lots of missing data for fossil taxa in supermatrix analyses (e.g., see Springer et al., 2001 for logic and advantages of molecular backbone/scaffolding). Bottom line again is that this is a poor argument and no excuse for eliminating fossil taxa that are less complete. This whole section on pages 12-14 will all have to change depending on how my above comments are addressed.

Discussion, page 15, line 245: "robust mandibles" robust in what dimensions? Be specific

Discussion, page 15, lines 250-252: This comparison between Ouranopithecus and Anadoluvius needs to be provided in a figure somewhere at the very least, and ideally some sort of measurement or visualization to demonstrate the verticality should be provided.

Discussion, page 16, lines 280-281: "broad agreement that Ouranopithecus is a likely hominine..." Agreed, so why the need for the totally unconvincing and methodologically poor phylogenetic analysis provided in this paper? That is why I think previous analyses can be cited and the current analysis could just be removed.

Discussion of phylogenetic analysis pages 16-17: Will all have to be revised or removed given my above comments.

Supplementary Materials

Page 3, lines 54-57: I don't understand what you mean by the stating that the premaxilla is short and vertically oriented. Relative to what? Can you illustrate this or point to it in a figure? Figs. S1-S2 don't illustrate this and from the reconstruction of the partial face in Fig.S S4, the premaxilla sticks out quite a bit. A comparison photo with something else or some sort of indication of what this feature is measuring would be helpful.

Also, the incisor orientation is much clearer on the right side than the left, which appears to have the incisors more posterior and closer to the canine than the right side, which actually preserves the contact between I1 and C1. And CO 2100/2800 also seems to suggest the I1 is quite a bit anterior to the C1. So maybe rephrase or explain this further? Again, it would be better/clearer if this was quantified. See my suggestions in the main text for a canine to prosthion measurement or something like that.

Page 4, lines 67-68: Similar comment as above. Please illustrate or measure the relatively short nasoalveolar clivus. You name a bunch of taxa for comparison here...Can the relatively short clivus in CO-205 be demonstrated with a figure or a boxplot with these taxa for comparison? Maybe combine such a figure and address this feature and the short premaxilla described above?

Figure S1: I certainly agree that the positioning of the incisors on the right panel (original publication) seems off. And I also agree that the position of the incisors in the reconstruction on the left panel seems much more reasonable relative to the midline. However, how do you know that the incisors are "in alignment with the canines" mesiodistally? From the picture on the left, I don't see any bones and teeth in contact that would confirm that position. This seems unclear to me as to why they are positioned so far posteriorly. On the right side of the maxilla, the I1 alveolus appears to be a bit further anterior than the left I1 is relative to the left C1, where there is no contact. I agree that the lower face/premax is still probably short, but the right side appears more accurate than the left side here. Please explain further or clarify and, again, a measurement here would be helpful.

Differential diagnosis, Page 14, lines 232-235: A number of the features listed, particularly for the face, differentiate *Anadoluvius* females from *Ouranopithecus* females only, correct? If a male face isn't known yet for *Anadoluvius*, some features can only be compared to other females. Sex-specific comparisons should be made clear here since males might be distinguished in different ways for these highly dimorphic taxa.

Table S3, PCA: First, the raw measurements for all specimens that were included in the PCA should be provided. This could be organized and uploaded in an Excel spreadsheet. Second, as mentioned above, the analysis needs to be re-run with size-adjusted variables either in addition to or instead of the raw measurement analysis here. I would suggest calculating a GMean for these 9 variables and then divide each variable by the GMean to express everything in relative terms. Third, Pan and Gorilla need to be included. Fourth, in table 3c (Loadings), the measurement abbreviations being used need to be defined in the caption somewhere. And finally, also in Table 3c or a separate table, correlations should be run between size (as defined by the GMean for each specimen or some other size proxy

variable) and the PC axes as well. We need to know how much size is driving the PCs.

Figure S5: These graphs are almost impossible to read. The text is too fuzzy and too small. Furthermore, these provide very little information about variation among taxa. Also, why aren't Pan and Gorilla plotted in the top (lower canine) plot? Each of these indices should be pulled out (c/p3, c/p4, c/m1, and c/m2 for both uppers and lowers) and plotted as separate box and whisker plots so we can see variation within each taxon. Furthermore, some basic statistical tests should be conducted, such as an ANOVA with post-hoc comparisons to look for significant differences between taxa. And again, comparative samples of Pan and Gorilla should be added to the statistical comparisons.

Figure S6: These box plots are nice, but it seems to me a couple more should be added, and again Pan and Gorilla need to be included so that we can judge typical variation within known hominine taxa for these features as well as test for significant differences (ANOVA with post-hoc comparisons). First, it seems like not just p4, but also P4 from the figures is obviously larger and should be plotted. And it's not just p4/P4 SIZE, but also SHAPE....Anadoluvius seems to have a much more MD elongated P4, making its outline much more rounded compared to Ouranopithecus, which has primitive looking/very buccolingually broad premolars. So the shape could be plotted there for comparison. Finally, the upper M3 in Anadoluvius also looks elongated compared to Ouranopithecus. Can molar proportions relative to each other be provided in boxplots as well? IT seems to me the relative size of M3 compared to M1 or M2 should probably separate the taxa as well.

Figure S7: Are these Raw measurements? They should again be size-adjusted by a GMean or other size correlate of some sort. Also, again....please plot these measures individually as box and whisker plots to make them easier to read. Extant hominines should be included in these plots as well, along with statistical tests (ANOVA with post-hoc comparisons). I understand that the taxa plotted here are roughly similar in size, but it is bad practice to be plotting raw measurements that haven't been size-corrected, particularly when you size-correct other measurements such as mandibular breadth. In addition, if you include the extant hominines (which you should), then size-correction becomes necessary anyway.

Phylogenetic Analysis/Matrix: Please provide any updated matrix as a nexus file, text file, or even an Excel file rather than embedded here.

Again, my other minor comments are provided on the annotated PDFs and should be addressed as well.

I look forward to seeing a revised draft of this interesting manuscript.

**Title: A new hominine from Turkey and the radiation of late Miocene circum-**
**Mediterranean apes..**

Large apes have been known from late Miocene deposits in the Balkans since 1951. Most
fossils come from several sites in Macedonia (Greece), with scattered specimens from Attica
(Greece) and Bulgaria. These samples, assigned to *Ouranopithecus* and *Graecopithecus*, are
variously attributed to stem hominids (great apes and humans), hominines (African apes and

humans) or hominins. In Anatolia, *Ankarapithecus* from Sinap is probably a pongine, while a
small sample from Çorakyerler has been attributed to *Ouranopithecus*. A new partial cranium
from Çorakyerler, described here, differs from *Ouranopithecus*, *Graecopithecus*, and
*Ankarapithecus*, indicating that the fossils represent a new genus. *Anadoluvius* is
distinguished from other eastern Mediterranean apes in the palate, face, neurocranium,
mandible, and in dental proportions and morphology. *Ouranopithecus*, *Anadoluvius* and
*Graecopithecus* may be members of an evolving lineage, with the new data from Çorakyerler
further supporting the hominine affinities of these taxa. Hominines were more diverse in the
late Miocene of the eastern Mediterranean than previously understood, with a known range
from at least 9.6 to 7.2 million years ago.

Late Miocene apes from the eastern Mediterranean were first described from Pyrgos
Vassilissis, in Athens (Attica, Greece), based on a single battered mandible attributed to the
genus *Graecopithecus* (1-3). A single tooth from Azmaka in Bulgaria was recently attributed
to *Graecopithecus* (3,4). Since then, many fossils of thickly enameled late Miocene apes have
been described from Macedonia (Northern Greece) and Anatolia, attributed to the genus
*Ouranopithecus*, the affinities of which have long been debated (5-16) (fig. 1.) There is also
debate regarding the distinction between the two, with *Ouranopithecus* occasionally
synonymized with *Graecopithecus* (17) (see Supplementary Information for more details.)
The prevailing view until recently was that, regardless of taxonomic status, *Graecopithecus*
and *Ouranopithecus* were almost indistinguishable. A recent re-analysis of the
*Graecopithecus* mandible and premolar has called this into question (3, see also 7).

Fig. 1. Satellite image of the eastern Mediterranean and the Anatolian peninsula with hominid localities. Inset: List of hominid localities, their ages, and a summary of the known fossils.

For some time, the only late Miocene ape from Anatolia was *Ankarapithecus*, which is alternatively described as a stem hominid or a pongine (18-20), but not a hominine. It is easily distinguished from the Balkan apes. In 2007, a new species of *Ouranopithecus* was described from Çorakyerler in central Anatolia (21). Since then, thousands of vertebrate fossils have been recovered at Çorakyerler, including a well-preserved ape partial cranium (22) (fig. 2.) The *O. turkae* holotype, a fragmented palate, was originally distinguished from *O. macedoniensis* in its shorter premaxilla, narrower palate, morphologically similar (homomorphic) upper premolars (as opposed to P3 being more triangular than P4), smaller male canines and possibly larger size (21). However, recovery of the new cranium and our reanalysis of the published material requires a reassessment of this conclusion and justifies the naming of a new genus of Miocene hominine. This conclusion, along with the recent reassessment of the Pyrgos Vassilissis and Azmaka specimens reveals a hitherto unappreciated diversity of late Miocene apes in the eastern Mediterranean.

**Fig. 2.** CO 2100/2800. From left to right, palatal, right lateral and anterior views.

Systematic paleontology

Order Primates Linnaeus, 1758

Infraorder Catarrhini Geoffroy, 1812

Superfamily Hominoidea Gray, 1825

Family Hominidae Gray, 1825

Subfamily Homininae Gray 1825

*Anadoluvius* gen. nov.

*Synonymy*

*Ouranopithecus* Bonis and Melentis: Güleç et al. 2007

*Type species*

*Anadoluvius turkae* comb. nov. Sevim Erol et al. 2021

*Etymology*

Anadolu is the modern Turkish word for Anatolia and Anatolian.

*Holotype*

CO-205, a fragmented but largely complete palate with LI¹-M³ and RC-M² (Fig. S1)

*Paratypes*
CO-300 (RM₂); CO-305 (mandibular fragment with RC-M₁); CO-710 (mandibular
fragment with RP₃-M₂); CO-2100 (RI¹); CO-2800 (partial cranium with RC-M², portions of
the right maxilla, maxillary frontal processes, frontal maxillary processes and most of the
frontal bone) (Figs. S 2-4.) The revised diagnosis for this new taxon appears in the

supplementary information. *Anadoluvius* is compared with other fossil hominids and *Ekembo*
in table S.4 of the SI.

**Results**

The CO-2100/2800 partial cranium (figure 2) was recovered with some crushing and
displacement of several broken pieces (see SI for restoration details and detailed description.)

To summarize, the frontal bone is nearly intact, missing only portions within the temporal
fossa and approaching bregma. This distinguishes it from the most complete facial specimen
of *Ouranopithecus* (XIR-1), which is broken just beyond the superior orbital margins. The
premaxillary alveolar clivus is preserved from prosthion to the base of the nasal aperture. In
figure 2 this is partly obscured by wax and due to repositioning in the reconstruction.

Distortion in the maxilla between the alveolar process and the nasal pillar was corrected,
leading to a re-orientation of the premaxilla, shifting it slightly more posteriorly and more
horizontally (see SI for details). Alignment with the maxilla as well as a constraint imposed
by the length of the I¹ root positions the base of the nasal aperture superior to and
overlapping the palatine process of the maxilla, producing a stepped subnasal fossa. Though

damaged, the position and orientation of the premaxilla is better preserved in CO-2100/2800
than in CO-205 and *Ouranopithecus* (XIR-1 and RPI 128), confirming previous
interpretations of a stepped and overlapping morphology in these specimens (6, 7, 23-25).

Areas of distortion that remain after reconstruction of CO-2100/2800 are apparent in
figure 2. Most obvious is the right alveolar process, which is crushed and pushed superiorly
and medially into the infratemporal fossa (lateral view.) Plastic deformation between the
nasal pillar and the maxillary alveolar process shifts the nasal margin laterally, distorting the
breadth of the nasal aperture (anterior view). The nasal aperture is further artificially enlarged
by distortion between the maxillary and zygomatic portions of the inferior orbital margin on
the left side. These distortions have been corrected in the reconstruction, which was informed
by preserved contours, matching conjoining breaks, and comparisons with *Ouranopithecus*
and CO-205 (figures S 2 and 4; see SI for details).

The premaxilla of *Anadoluvius* is short and vertical compared with *Pan*, *Pongo*, and
australopithecines, and is most like *Gorilla*, being relatively short in the alveolar portion but
expanded nasally to overlap with the palatine process of the maxilla. The incisor alveoli are
positioned along the mesial transverse plane of the canine crowns (fig. S1.)

In two specimens of *Ouranopithecus* (RPI 128 and XIR 1) the upper incisors are well
anterior to the canines (fig. S2.) In NKT 89 the premaxilla is severely damaged, but the
posterior edge of the lateral incisor appears to be aligned with the anterior transverse plane of
the canines, a position most like *Anadoluvius*. The frontal bone of *Anadoluvius* differs
strongly from that of *Ouranopithecus* in the smooth biconvex squama of the former,
contrasting with a broad concavity above glabella in the latter. As well, the superior orbital
margins of *Ouranopithecus* broad, rounded and slightly projecting while they are sharp
and flat in *Anadoluvius*.

The left alveolar process of the CO-305 mandible is well preserved from P₃ to M₁ while
 the right corpus lacks most of the buccal cortical bone. There is a crack between the I₂ and
 canine on the right side, but the symphysis is in close to proper position relative to the right
 corpus. The lingual surface of the symphysis is similar in inclination to male
 *Ouranopithecus*, more inclined than in female *Ouranopithecus*. More detailed descriptions of
 all the specimens can be found in the SI.

Figure 3 shows the root and root canal morphology of CO-300, the male mandible of
 *Anadoluvius* (see methods and SI for segmentation details). Unlike *Ouranopithecus*, the
 distal roots of P₃ to M₁ in *Anadoluvius* and *Graecopithecus* (3) are single fused roots with 2
 root canals (fig. 3 and table 1).

	CO-300	Graecopithecus	Ouranopithecus	Sahelanthropus
C	1□ (1)	1□ (1)		
P□	1□M + 1□D (1)	1□M+1□D (1)	1□M+2□D (4)	11M + 12D (2)
P□	1□M + 1□D (1)	1□M+1□D (1)	1□M+2□D (2); 2□M+2□D (2)	12M + 12D (2)
M□	2□M + 1□D (1)	2□M+1□- 2 D (1)	2□M+1□D (4)	22M + 12D (2)
M□		1-2□M+1□D (1)	1□M+1□D (2); 2□M+1□D (3)	22M + 12D (2)
M□		1□M+1□D (1)	1□M+1□D (4); 2□M+1□D (1)	
Ref.	This study	(3)	(3)	(72)

Table 1. Lower tooth root formula in *Anadoluvius*, *Graecopithecus*, *Ouranopithecus* and
 *Sahelanthropus*. Explanation of the formula: 22M + 12D is 2 mesial roots with 2 separate
 root canals plus 1 distal root with 2 separate root canals. Multiple formulae in
 *Ouranopithecus* represents variability in that taxon.

Fig. 3. 3-D reconstruction of the left P₃ to M₁ of CO 300, showing the root, root canal and pulp chamber configurations. See table 1 for a comparison of root formulae. Scale =10 mm.

 **Fig 4.** Lower M₂ relative enamel thickness in selected hominoids. The thin blue line marks the
 value for *Anadoluvius*. The blue star represents RET in an M₁ of *Ouranopithecus* (RET for M₂
 is not available in *Ouranopithecus*).

Figure 4 illustrates ranges of variation in relative enamel thickness in Miocene, Plio-
 Pleistocene and living hominoids. *Anadoluvius* has thicker enamel than most Miocene apes,
 falling at the upper end of the range in *Afropithecus*. Its RET is lower than the RET of RPI
 641, an M₁ of *Ouranopithecus*. M₁ normally has thinner enamel than M₂, so this value is

likely to underrepresent RET in *Ouranopithecus* M₂. *Anadoluvius* falls well above the ranges
in extant hominids and within the *A. afarensis* and *A. africanus* 25-75% quartiles.

A principal components analysis based on mandibular measurements available for
*Graecopithecus*, *Ouranopithecus* and *Anadoluvius* is presented in figure 5a. PC1 and PC2
account for 83 and 14% of the variance respectively. PC1 is driven primarily by corpus
breadth and PC2 by symphyseal-M₁-M₂ distance and mandibular arch breadth (measurements
defined in the Methods section). Male and female *Ouranopithecus* are separated from each
other and from NKT 21, *Anadoluvius* and *Graecopithecus*. In this and in the subsequent
analyses NKT 21 was considered separately as there is a question of its inclusion in
*Ouranopithecus macedoniensis* (8).

Figure 5b shows the relationship of lower canine to M₂ size in a diversity of fossil and
extant apes. Canines are small in the Balkan/Anatolian sample compared with other fossil
and extant apes including *Ardipithecus*, being more consistent with the ranges in
*Australopithecus*. The ratio of lower canine to M₂ size in *Anadoluvius* is lower than in any
male and most females except *Australopithecus*. To investigate whether this reflects small
canines or large molars, both were compared to mandibular arch breadth at M₂ in *Pan*
*troglydites* males, which are not considered to be megadont. The results show that all the
Balkan/Anatolian specimens have relatively large M₂'s compared with *Pan* males (figure
5c), while relative lower canine size in Ravin de la Pluie males is indistinguishable from *Pan*
males (figure 5d). The single *Anadoluvius* canine falls below the 25% quartile for *Pan* males
but within the range, NKT 21 is at the extreme small end of the *Pan* range and Ravin females
are below the *Pan* male range (figure 5d).

Fig. 5. PCA and canine size comparisons. A) PCA based on mandibular measurements. Sexes and individual samples are well

separated along both axes. B) Box plots of male and female lower canine size (cervical dimensions), with Balkan taxa falling

among Pliocene hominins. C) Box plots showing the relatively enlarged M_2 in the fossil samples compared with male *Pan*. D)

Box plots show no difference in relative canine size between *Ouranopithecus* and *Pan* males and somewhat smaller canines in

CO-305 and NKT 21. See text.

The failure to distinguish *Ouranopithecus* and *Pan* males in relative lower canine size
contrasts with the conventional view that *Ouranopithecus* has relatively small canines (9, 25;
but see (26)). Canine size in this analysis is measured at the cervix and does not account for
canine crown height. An evaluation of upper and lower male canine crown height in the
Balkan/Anatolian samples is hampered by the paucity of complete, unworn crowns.

Figure 6 summarizes the analysis of relative mandibular robusticity and selected dental
size ratios. *Anadoluvius* is distinct from *Graecopithecus* in all mandibular and dental ratios. It
falls beyond the range of variation of *Ouranopithecus* in relative corpus breadth at M₁-M₂, P₄
length, M₂ size and M₂ size relative to mandibular corpus breadth (figure 6a,d,e,f).
*Anadoluvius* is outside the 25-75% quartile for corpus breadth at P₄ and molar-symphysis
distance (figure 6b,c). Interestingly, NKT 21 falls outside the *Ouranopithecus* range in
relative P₄ length and M₂ size (figure 6d,e). It has a relatively short symphyseal-molar
distance, at the 25% quartile for the Ravin sample, and a relatively robust mandible at M₁-
M₂, at the 75% quartile for *Ouranopithecus* (figure 6a,c). In *Graecopithecus* the symphysis is
closest to the molars, just barely in the range of the Ravin sample. In summary, in most
quantitative comparisons *Anadoluvius* is distinguished from *Ouranopithecus* and
*Graecopithecus*. Additional quantitative analysis is presented in table S4 and figures S 5-7.

The results of a cladistic analysis using a data matrix of 112 characters selected to
minimize missing data are presented in figure 7. Debate on the influence of missing data in
morphological data sets is summarized in (27-30.) There are many advantages to including as
many fossil taxa as possible in phylogeny reconstruction, but most of the analysis of this
issue is based on data sets that combine morphological and molecular data, in which a large
proportion of the data are molecular and many of the OTU's are extant. In addition, in most

Fig. 6. Box plots illustrating relative mandibular robusticity and dental size. In 5a-e the dimension is standardized by the geometric mean. In 5f M_2 size is compared with corpus breadth to include *Nakalipithecus*. A) Mandibular corpus breadth between M_1 and M_2 , showing a more robust mandible in *Anadoluvius*. B) Mandibular corpus breadth at P_4 , with a more gracile mandible at this level in *Graecopithecus*. C) Horizontal distance between the symphysis and the corpus between M_1 and M_2 , with *Anadoluvius* just above the 75% quartile for the combined sex sample of *Ouranopithecus* and *Graecopithecus* at the extreme low end of the range of variation. D and E) P_4 length and M_2 relative size. All three non-Ravin de la Pluie samples have relatively larger teeth, especially *Anadoluvius* and *Graecopithecus*. F) M_2 size relative to mandibular corpus breadth. *Graecopithecus* has an enlarged M_2 while the male *Anadoluvius* specimen falls below the range of variation of male *Ouranopithecus*.

cases the fossils included are relatively complete. In the case of data sets dominated by fossil

taxa, inclusion of taxa that result in high amounts of missing data have a strongly negative

impact on congruence (27, 30) Therefore, in this analysis the data matrix is based on

characters present in *Anadoluvius*, the focus of this research.

Fig. 7. Consensus cladogram based on two MPT's. See text.

The consensus tree (figure 7) is based on two most parsimonious trees (MPT's). The

MPT's differ in positioning the *Anadoluvius-Ouranopithecus* clade as sister to either crown

hominines or dryopithecins (SI figure S 8). Synapomorphies are mapped onto both MPT's

and Bremer support values are mapped onto the consensus cladogram in SI figure S 8.

Bremer support values are high for the hominine clade that includes dryopithecins and the

Ouranopithecus-Anadoluvius clade, and the pongine clade that includes *Sivapithecus* and

Ankarapithecus. These results are broadly consistent with many but not all previous analyses

(see discussion). Noteworthy is the recovery in these data dominated by fossils of the two

known clades of extant hominids, pongines and hominines, and the recovery of the widely
recognized *Pan*-hominin clade.

Discussion

*Ouranopithecus*, *Graecopithecus* and *Anadoluvius* share a suite of derived characters of
the jaws and dentition that support their status as sister taxa. The core attributes are thickly
enameled molars, robust mandibles, small canines relative to molar size and large size.

However, among these taxa there is diversity. *Graecopithecus* is distinguished from
*Ouranopithecus* in its relatively large M_2 compared with both the M_1 and corpus breadth, its
more vertical mandibular symphysis and in details of root morphology (3, 7, 8, 12, 13).

*Anadoluvius* has the same lower dental root formula (canine to M_1) as *Graecopithecus* and
both differ from *Ouranopithecus*. The frontal bone between the superior orbital margin and

the anterior temporal line is preserved in XIR-1 (*Ouranopithecus*), which is sufficient to

show that it was more vertically oriented than in CO-2100/2800. *Anadoluvius* is further
distinguished from *Ouranopithecus* and other non-hominin hominines in having
mesiodistally shorter canines (as in *Graecopithecus* based on canine root size) that lack
mesial grooves and lingual cingula (unknown for *Graecopithecus*.)

In quantitative attributes, figure 5a illustrates the overall distinctiveness of *Anadoluvius*
compared with Balkan apes. More specifically, while the *Anadoluvius* mandible CO-300/305
is attributed to a male based on canine morphology (see also 21), it is distinguished from
male *Ouranopithecus* in many metric comparisons (canine- M_2 ratio, canine/mandibular
breadth, relative mandibular breadth, symphyseal-molar distance, relative P_4 length and size,
and relative P_4 , M_1 and M_2 size (figures 5-6 and figures S 5-7.) *Anadoluvius* is distinguished
from male *Ouranopithecus* in canine size relative to postcanine tooth size at every tooth

position for both upper and lower tooth rows (figure S 5). *Anadoluvius* is distinguished from
*Graecopithecus* in relative mandibular breadth, symphyseal-molar distance, P₄ length, M₂
relative size and M₂ size relative to corpus breadth (figures 5-6). In addition, the mandibular
arch is wider in and relative mandibular breadth larger at every dental position in
*Anadoluvius* (figure S 7).

*Ouranopithecus* and *Anadoluvius* lack shared derived characters of the pongines
(airorhynchy, greatly elongated premaxilla substantially or completely overlapping the
maxillary palatine process, expanded zygoma, tall orbits, narrow interorbital space, reduced
or absent ethmoidal frontal sinus, circumorbital costae.) There is no evidence for their
inclusion in Ponginae.

*Ouranopithecus* and *Anadoluvius* share with dryopithecins [European middle and late
Miocene apes with affinities to *Dryopithecus* (6,7)] a series of characters found among
hominines. These include a ventrally rotated palate, a stepped subnasal fossa, broad, flat
nasal aperture base, short nasal bones, nasal aperture apex superior to the infraorbital
margins, robust lateral orbital pillars, frontal sinus expanded below nasion, incipient
supraorbital torus, more horizontal frontal squama (6-8). The phylogenetic significance of
some of these shared attributes are disputed, particularly concerning the dryopithecins.

However, there is broad agreement that *Ouranopithecus* shares enough derived characters
with hominines to warrant inclusion in that group (6-16).

The phylogenetic analysis presented here is the most comprehensive one focused on
eastern Mediterranean hominids. Unfortunately, *Graecopithecus* could not be included in the
cladistic analysis as only 2 of the 112 characters in the matrix are present in the two known
specimens. The results regarding *Ouranopithecus* and its sister, *Anadoluvius*, are consistent
with many previous analyses (6-16, 23-25, 31-33). The results of the current analysis are

robust in terms of the number of synapomorphies and Bremer support values for many clades
(see figure S 8). The results also recover the widely accepted relations among crown
hominids and hominines and relations within all fossil clades. The Bremer support for the
hominine clade as defined here is high (six), and very good (four) for the pongine clade.
European and eastern Mediterranean apes are classified as hominine in both MPT's. In the
consensus cladogram the relations among the dryopithecins, the *Ouranopithecus*-
*Anadoluvius* clade and crown hominines are unresolved, with the *Ouranopithecus*-
*Anadoluvius* clade falling among the dryopithecins in one MPT and the crown hominine
clade in the other (figure S 8.)

Other taxa have been linked with *Ouranopithecus* or hominines. Three fossil apes from
Africa, *Nakalipithecus*, *Samburupithecus* and *Chororapithecus*, overlap in time with
*Ouranopithecus*, *Graecopithecus* and *Anadoluvius* (34-36). *Nakalipithecus* has been
identified as potentially ancestral to *Ouranopithecus* (35). However, *Nakalipithecus* differs in
many details of dental morphology from *Anadoluvius* and *Ouranopithecus* (see the
differential diagnosis in the supplemental information.) It has been hypothesized that both
*Chororapithecus* and *Ouranopithecus* have phylogenetic affinities with gorillas (16, 35-37).
*Chororapithecus* is readily distinguished from both *Anadoluvius* and *Ouranopithecus* (see
differential diagnosis), and its affinity with gorillas has been questioned (6, 38). (34) claim
that *Samburupithecus* has phylogenetic affinities with African apes and humans, though this
conclusion has been challenged by (6, 39), who conclude that *Samburupithecus* is not a
hominid but instead a vestige of the early Miocene proconsuloid radiation. Regardless of the
relations of *Samburupithecus*, it is clearly distinct from *Anadoluvius* (see differential
diagnosis.)

*Diversity and paleobiogeography*

A comprehensive review of the taxonomy and phylogeny of late Miocene apes is
needed, given recent discoveries and reinterpretations (8). Here we focus on the diversity and
paleobiogeographic implications of *Anadoluvius*. *Anadoluvius* and *Ouranopithecus* share
many attributes with other European middle and late Miocene hominids that distinguish them
from late Miocene ape fossils from Africa and the broadly contemporaneous pongines
*Sivapithecus*, *Ankarapithecus* and *Khoratpithecus* (table S5) (7). *Ouranopithecus* from Ravin
de la Pluie and Xirochori are dated to 9.6 and 9.3 Ma respectively (40). *Ouranopithecus* from
Nikiti 1 is dated to between 8.5-9 Ma (40). The Nikiti mandible and maxilla are distinct from
*Ouranopithecus* from the more northern Macedonian sites and may represent a different
taxon (7). This possibility, which needs further study, is interesting in terms of regional
evolution as Nikiti 1 is likely to be close in age to Çorakyerler (22, 40) (fig. 1.) Nikiti,
*Anadoluvius* and *Graecopithecus* are distinguished from *Ouranopithecus* in our PCA and in
having a greater degree of canine reduction, elongated P₄ and large molars relative to
mandibular corpus size. *Anadoluvius* and *Graecopithecus* are distinguished from
*Ouranopithecus* in root morphology (unknown in NKT 21.) NKT 21 and *Graecopithecus* have
inferior transverse tori positioned posterior to the mesial edge of the M₁ (unknown in
*Anadoluvius*), while the torus is anterior to the M₁ in *Ouranopithecus*. *Graecopithecus*, which
is considerably younger (~7.2 Ma), has been shown to differ from *Ouranopithecus* in
morphology that replicates differences between late Miocene apes and early hominins, such as
reduced relative canine size and premolar root morphology (3, 41). However, the Nikiti
specimens have not previously been included in these comparisons.

Quantitative and qualitative comparisons reveal many differences among the
Balkan/Anatolian taxa, indicative of a greater diversity of late Miocene eastern

Mediterranean hominids than previously recognized. Hominines appear to have been present
and diverse for millions of years in the late Miocene of Europe. Based on the large number of
qualitative and quantitative differences among the samples from Macedonia, Attica and
Anatolia, we conclude that they represent at least three distinct hominine genera,
*Ouranopithecus*, *Graecopithecus* and *Anadoluvius*. The diversity of hominines in the eastern
Mediterranean mirrors that among australopithecines in the Plio-Pleistocene hominin record
in Africa. In the phylogenetic analysis presented here the Balkan/Anatolian taxa are the sister
clade to crown hominines.  The fact that dryopithecins are also classified as hominines
suggests that there was *in situ* evolution of thickly enameled late Miocene eastern
Mediterranean hominines from precursors in central and western Europe. *Pierolapithecus*,
*Anoiapithecus*, *Dryopithecus*, *Hispanopithecus* and *Rudapithecus* all share attributes with
extant hominines and are distinguished from pongines such as *Sivapithecus* and
*Ankarapithecus* (6,7,23,24) (but see (42) for an alternative view) (figure 7.) A clade that
includes both thinly and thickly enameled taxa, in this case the dryopithecins and the
Balkan/Anatolian apes, has a parallel in Africa with *Ardipithecus* and australopithecines and
with *Pan* and hominines.

The parallel evolution of thickly and thinly enameled members of a clade in Africa
and Europe is not proof that the late Miocene European apes are all hominines, but it does
make this hypothesis, supported by the results of the cladistic analysis, plausible. It is
possible that the thinly enameled dryopithecins and the later occurring thickly enameled
Balkan/Anatolian hominines do not share an ancestor-descendant relationship and represent
separate dispersal events into Europe from Africa (e.g. 32), though this contrasts with the
results of the phylogenetic analysis presented here. While independent dispersal events are

possible, we regard the *in situ* European hypothesis as more likely and more parsimonious
given the current evidence.

Given the results of this analysis we propose the following taxonomy of the taxa
discussed in the text:

Table 2: Taxonomy of hominoids discussed in the text¹.

Hominoidea

Proconsulidae

Ekembo

Proconsul

Samburupithecus

Afropithecidae

Afropithecus

Morotopithecus

Hominidae

Griphopithecinae

Griphopithecus

Equatorius

Nacholapithecus

Kenyapithecus

Homininae

Dryopithecini

Dryopithecus

Pierolapithecus

Anoiapithecus

Danuvius

Hispanopithecus

Rudapithecus

Ouranopithecus

Graecopithecus

Anadoluvius

Gorillini

Gorilla

Hominini

Pan

Orrorin
Sahelanthropus
Ardipithecus
Australopithecus
Paranthropus

Ponginae

Pongo
Sivapithecus
Ankarapithecus
Khoratpithecus

Hominidae incertae cedis

Chororapithecus
Nakalipithecus
Lufengpithecus

Hominoidea incertae cedis

Oreopithecus

Notes: 1) Not all clades are named in this classification. There are at least two dryopithecine
clades as well as separate *Pan* and *Ardipithecus*/australopithecine clades. Multiple clades are
probably also contained within the Griphopithecinae and the Ponginae.

**Conclusions**

Eastern Mediterranean apes previously assigned to *Ouranopithecus* span at least 2.4
million years and represent multiple genera. Their thick enamel, robust jaws and, in the
younger samples from Nikiti 1, Çorakyerler and Pyrgos Vassilissis, evidence of canine
reduction, recalls the earliest African hominins. Unfortunately, the data are insufficient to
include *Orrorin* and *Sahelanthropus* in the phylogenetic analysis presented here. A
clarification of the significance of the similarities between the Balkan/Anatolian samples and
late Miocene African hominins will have to await further discoveries.

Eastern Mediterranean hominines may represent a terminal radiation arising from one or
more older hominines in Europe, analogous to the radiation of *Paranthropus* presumably
from an *Australopithecus*-like ancestor. Alternatively, given that European hominines most
closely resemble gorillas (7, 10-11, 16, 37), it is possible that this represents a radiation of
early members of the gorilla clade, mirroring the middle and late Miocene radiation of
pongines in Asia (43). It is also possible that European hominines represent terminal lineages
of successive dispersals of hominines from Africa (32), though there is no evidence of
multiple lineages of hominines between 13 and 10 Ma in Africa, and this hypothesis is not
supported by the results of the phylogenetic analysis presented here. Finally, some
researchers have focused attention on differences among eastern Mediterranean apes,
suggesting that multiple lineages are present, including the earliest known hominins [e.g. (3,
9).] A comprehensive phylogenetic analysis is underway to test these competing hypotheses.
Whatever the outcome, the sample of ape fossils from Çorakyerler demonstrates that great
ape diversity in the eastern Mediterranean is greater than previously believed and that
hominines had diversified into multiple taxa long before their first documented appearance in
Africa.

**Methods**

*Measurements:* Dental dimensions are standard mesiodistal and buccolingual
measurements with canine measurements crown maximum and perpendicular dimensions.
Mandibular breadth measurements were taken just below the margin of the alveolar process
to allow for comparisons with CO-300, which lacks the corpus base. The distance between
the symphysis and the M₁-M₂ interproximal space is a non-standard measurement that allows

for the comparison among all the specimens. It is the horizontal chord between the symphysis
and the M₁-M₂ interproximal space at the level of the alveolar margin, taken from high
resolution casts (dimensions verified by comparison with measurements on original
specimens). The geometric mean was calculated from the mandibular measurements and did
not include dental dimensions (table S2.) In cases in which mandibles were incomplete or
crushed, mandibular arch breadth between antimeres was estimated by doubling the distance
from the alveolar margin to the midline.

The diagnoses and descriptions here result from direct observation of original fossil
material by the first and second authors. All published specimens currently attributed to
*Ouranopithecus* were examined, measured and photographed. These observations were
compared with direct observations of all published European middle and late Miocene
hominids
(numerous repositories, see (23), see table S1 for explanation of the abbreviations), the type
and palate of *Ankarapithecus* (MTA), a mandible (MTA) and a sample of isolated teeth
(NHM) of *Griphopithecus* and *Kenyapithecus*, all published African Miocene catarrhines
(NMK, NHM, ENM), large portions of the sample of *Sivapithecus* (Harvard, AMNH) and the
IVPP sample of *Lufengpithecus* up to 1995. Comparisons with Pliocene hominins are based
on observations on original material of *Ardipithecus*, *Australopithecus* and *Paranthropus*
(NME, NMK) supplemented with high resolution casts of *Australopithecus* and other
Pliocene hominins. Data from original fossils were supplemented with high resolution casts
of *Morotopithecus* and *Proconsul* from Uganda, *Orrorin*, *Sahelanthropus* and published
descriptions and images of *Khoratpithecus*. Comparisons with extant hominoids are based on
data collected from the institutions listed in table S1. All specimens were measured by the
second author using digital calipers to the nearest 10th of a millimeter (Table S2).

CT images were processed by RMGM using well established methods. The complete
image stacks of each tooth were filtered using a three-dimensional median filter with a
kernel size of 1 followed by a mean of least variance filter with a kernel size of 1. The
filtered image stacks were imported into Avizo 6.3 (www.thermofisher.com) where the
enamel, dentine, pulp cavity, and cracks were segmented semi-automatically using the 3D
voxel value histogram and grayscale values. In Avizo 6.3, the teeth were reconstructed
from the segmentations as triangle-based surface models using the constrained smoothing
parameter.

Statistics: Quantitative analysis was carried out using PAST © (Hammer et al. 2001)
and © Excel for Microsoft 365. The PCA matrix was set at variance-covariance.
Eigenvalues and % variance, scores and loadings for each PC appear in table S 4.

Phylogeny: Two MPT's were recovered using TNT [Willi Hennig Society; ©
Goloboff & Catalano (2006). Cladistics. DOI 10.1111/cla. 12160]. Characters were
unweighted and unordered. Traditional search was used to generate 253 (maximum tree
buffer) replications. TNT was used to map synapomorphies and calculate Bremer support
using the DOBREM script and calculate a majority rule consensus tree (figure S8). The
characters and their states are listed in tables S5 and 6 and the data matrix in TNT format
appears in table S7.

[revised manuscript text omitted]

References:

1. Freyberg, B. von, Die Pikermifauna von tour la Reine (Attica). Annales géologiques des Pays

Helléniques., 1951. 3: p. 7–10.

2. Koenigswald, G.H.R., Ein Unterkiefer eines fossilen Hominoiden aus dem Unterpliozän

Griechenlands. Proc. Kon. Nederl. Akad. Wet. B, 1972. 75: p. 385-394.

3. Fuss, J. et al. Potential hominin affinities of *Graecopithecus* from the Late Miocene of Europe.

PLoS ONE 12 (e0177127) (2017).

4. Spassov, N. et al. A hominid tooth from Bulgaria: The last pre-human hominid of continental

Europe. J. Hum. Evol 62, 138-145 (2012).

5. Bonis, L. de, Bouvrain, G. & Melentis, J. Nouveaux restes de primates hominoïdes dans le

Vallésien de Macédoine (Grèce). C. R. Acad. Sci. D Paris 182, 379-382 (1975).

6. Begun, D. R. Fossil Record of Miocene Hominoids in Handbook of Paleoanthropology (eds.

Henke, W. & Tattersall, I.) 1261-1332 (Springer 2015).

7. Begun, D. R. *Dryopithecus*, Darwin, de Bonis and the European origin of the African apes and

human clade. Geodiversitas 31, 789–816 (2009).

8. Begun, D. R. et al. Hominin origins: New evidence from the eastern Mediterranean. Am. J. Phys.

Anthrop. 168, 15 (2019).

9. Bonis, L. de & Koufos, G. D. Our ancestors' ancestor: *Ouranopithecus* is a Greek link in human

ancestry. Evol. Anthropol. 3, 75-83 (1994).

10. Dean, D. & Delson, E. Second gorilla or third chimp? Nature 359, 676-677 (1992).

11. Pugh, K. D. Phylogenetic Analysis of Miocene Apes and Early Hominins Using Qualitative and

Quantitative Morphological Characters. Am. J. Phys. Anthropol. 168, 195-196 (2019).

12. Koufos, G. D. & Bonis, L. de The Late Miocene hominoids *Ouranopithecus* and *Graecopithecus*.

Implications about their relationships and taxonomy. Ann. Paleontol. 91, 227-240 (2005).

13. Finarelli, J. A. & Clyde, W.C. Reassessing hominoid phylogeny: evaluating congruence

the morphological and temporal data. *Paleobiology*. 30, 614-651(2004).

14. Folinsbee, K. & Brooks, D. Miocene hominoid biogeography: Pulses of dispersal and
differentiation. *J. Biogeog.* 34, 383-397 (2007).

15. Young, N. M. & MacLatchy, L. The phylogenetic position of *Morotopithecus*. *J. Hum. Evol.* 46,
163-184 (2004).

16. Ioannidou, M. et al. A new three-dimensional geometric morphometrics analysis of the
*Ouranopithecus macedoniensis* cranium (Late Miocene, Central Macedonia, Greece). *Am. J. Phys.*
*Anthropol.* 170, 295-307 (2019).

17. Martin, L. & Andrews, P. The phyletic position of *Graecopithecus freybergi* KOENIGSWALD.
*Cour. Forsch. Inst. Sencken.* 69, 25-40 (1984).

18. Begun, D.R. & Güleç, E. Restoration of the Type and Palate of *Ankarapithecus meteai*:
Taxonomic, Phylogenetic, and Functional Implications. *Am. J. Phys. Anthropol.* 105, 279-314 (1998).

19. Alpagut, B., et al. A new specimen of *Ankarapithecus meteai* from the Sinap Formation
of central Anatolia. *Nature* 382, 349-351 (1996).

20. Kappelman, J. et al. Hominoidea (Primates) in Geology and Paleontology of the Miocene Sinap
Formation, Turkey. (eds. Fortelius, M. et al.) 90-124 (Columbia University Press, 2003).

21. Güleç, E. et al. A new great ape from the late Miocene of Turkey. *Anthropol. Sci.* 115, 153-158
(2007).

22. Erol, A.S. et al. The late Miocene (early Turolian, MN 11) fauna of Çorakyerler (Turkey): New
collection of large mammals. 2017.

23. Begun, D. R. *European Hominoids in The Primate Fossil Record* (ed Hartwig, W.) 339-368
(Cambridge University Press, 2002).

24. Begun, D.R. Nargolwalla, M.C., Kordos, L., 2012. European Miocene hominids and the origin of
the African ape and human clade. *Evol Anthropol* 21, 10-23.

25. Bonis, L. de & Koufos, G. The face and mandible of *Ouranopithecus macedoniensis*:
description of new specimens and comparisons. *J. Hum. Evol.* 24, 469-491(1993).

- 26. Kelley, J., 2001. Phylogeny and sexually dimorphic characters: canine reduction in
*Ouranopithecus.*, in: Bonis, L.d., Koufos, G., Andrews, P. (Eds.), Hominoid Evolution and
Environmental Change in the Neogene of Europe. Volume 2. Phylogeny of the Neogene Hominoid
Primates of Eurasia. Cambridge University Press, Cambridge, pp. 269-283.
- 27. Pattinson, D.J., Thompson, R.S., Piotrowski, A.K., Asher, R.J., 2015. Phylogeny, Paleontology,
and Primates: Do Incomplete Fossils Bias the Tree of Life? *Systematic Biology* 64, 169-186.
- 28. Asher, R.J., Smith, M.R., Rankin, A., Emry, R.J., 2019. Congruence, fossils and the evolutionary
tree of rodents and lagomorphs. *Royal Society open science* 6, 190387-190387.
- 29. Guillerme, T., Cooper, N., 2016. Effects of missing data on topological inference using a Total
Evidence approach. *Molecular Phylogenetics and Evolution* 94, 146-158.
- 30. Sansom, R.S., 2015. Bias and Sensitivity in the Placement of Fossil Taxa Resulting from
Interpretations of Missing Data. *Systematic Biology* 64, 256-266.
- 31. Emonet, E.-G., Tafforeau, P., Chaimanee, Y., Guy, F., de Bonis, L., Koufos, G., Jaeger, J.-J.,
2012. Three-dimensional analysis of mandibular dental root morphology in hominoids. *Journal of*
*Human Evolution* 62, 146-154.
- 32. Kunitatsu, Y., Nakatsukasa, M., Sawada, Y., Sakai, T., Hyodo, M., Hyodo, H., Itaya, T.,
Nakaya, H., Saegusa, H., Mazurier, A., Saneyoshi, M., Tsujikawa, H., Yamamoto, A., Mbua, E.,
2007. A new Late Miocene great ape from Kenya and its implications for the origins of African great
apes and humans. *Proceedings of the National Academy of Sciences* 104, 19220-19225.
- 33. Nengo, I., Tafforeau, P., Gilbert, C.C., Fleagle, J.G., Miller, E.R., Feibel, C., Fox, D.L., Feinberg,
565 J., Pugh, K.D., Berruyer, C., Mana, S., Engle, Z., Spoor, F., 2017. New infant cranium from the
566 African Miocene sheds light on ape evolution. *Nature* 548, 169-174.
- 34. Ishida, H. & Pickford, M. A new late Miocene hominoid from Kenya: *Samburupithecus*
*kiptalami* gen. et sp. nov. *C. R. Acad. Sci. Paris.* 325, 823-829(1997).
- 35. Kunitatsu, Y. et al. A new Late Miocene great ape from Kenya and its implications for the
origins of African great apes and humans. *Proc. Natl Acad. Sci.* 104, 19220-19225 (2007).

- 36. Suwa, G. et al. A new species of great ape from the late Miocene epoch in Ethiopia.
Nature. 448, 921-924 (2007).
- 37. Begun, D. R. Very old hominoid divergence dates based on paleontological and molecular data.
574 J. Vert. Paleontol. 87, 2015.
- 38. Harrison, T., 2010. Chapter 24. Dendropithecoidea, Proconsuloidea, and Hominoidea (Catarrhini,
Primates), in: Werdelin, L., Sanders, W.J. (Eds.), Cenozoic Mammals of Africa. University of
California Press, Berkeley, pp. 429-469.
- 39. Olejniczak, A.J., Begun, D.R., Mbua, E., Hublin, J.J., 2009. New evidence on the phylogenetic
position of *Samburupithecus*. Am. J. Phys. Anthropol. S 48, 202.
- 40. Koufos, G. D. The Neogene mammal localities of Greece: faunas, chronology and
biostratigraphy. Hell. J. Geosci. 41, 183-214 2006.
- 41. Böhme, M. et al. Messinian age and savannah environment of the possible hominin
*Graecopithecus* from Europe. PLoS ONE 12, e0177347 (2017).
- 42. Alba, D. M. Fossil apes from the Vallès-Penedès basin. Evol. Anthropol. 21, 254-269 (2012).
- 43. Kelley, J., 2002. The Hominoid Radiation in Asia, in: Hartwig, W. (Ed.), The Primate Fossil
Record. Cambridge University Press., Cambridge, pp. 369-384.
- 44. Cameron, D. W. The taxonomic status of *Graecopithecus*. Primates. 38, 293-302(1997).
- 45. Groves, C. P. A theory of Primate and Human Evolution. (Clarendon Press, 1989).
- 46. Koufos, G. D. & Bonis, L. de The Late Miocene hominoids *Ouranopithecus* and *Graecopithecus*.
Implications about their relationships and taxonomy. Ann. Paleontol. 91, 227-240 (2005).
- 47. Koufos, G. D. History, stratigraphy and fossiliferous sites. Geobios. 49, 3-10 2016.
- 48. Kaya, F. et al. Magnetostratigraphy and paleoecology of the hominid-bearing locality
Çorakyerler, Tuglu Formation (Çankiri Basin, Central Anatolia). J. Vert. Paleontol. 36, DOI:
10.1080/02724634.2015.1071710 (2016).
- 49. Kaymakçı, N. et al. Neogene tectonic development of the Çankırı basin (central
Anatolia, Türkiye). TPJD Bülteni. 13, 27-56 (2001).

- 50. Mazzini, I. et al. Palaeoenvironmental and chronological constraints on the Tuğlu Formation
(Çankırı Basin, Central Anatolia, Turkey). *Turk. J. Earth Sci.* 22, 747-777(2013).
- 51. Şen, Ş. et al. Mammalian biochronology of Neogene deposits and its correlation with the
lithostratigraphy in the Çankırı-Çorum basin, central Anatolia, Turkey. *Eclog.*
*Geolog. Helv.* 91, 307-320 (1998).
- 52. Sevin, M. & Uğuz, M.F. 1/100.000 scaled geological map sheets of Turkey (Çankırı- G31).
Mineral Research and Exploration Directorate of Turkey: Ankara (2011).
- 53. Öcal H., Turhan N. & Göktaş F. Geological maps of Turkey in 1:100000 scale: Çankırı G31
sheet. Mineral Research and Exploration Directorate of Turkey, Ankara 147, 32 pp. (2011).
- 54. Sickenberg, O. et al. Die Gliederung des höheren Jungertertiärs und Altquartärs in der
Türkei nach Vertebraten und ihre Bedeutung für die internationale Neogen-Stratigraphie.
*Geol. Jb. B(15)*, 1-167 (1975).
- 55. Heissig, K. Rhinocerotidae aus dem Jungtertiär Anatoliens. *Geol. Jb. B(15)*, 145- 151 (1975).
- 56. Geraads, D. Large Mammals from the late Miocene of Çorakyerler, Çankırı, Turkey.
*Acta Zoolog Bulgar.* 65, 381-390 (2013).
- 57. Köhler, M. Funktionsmorphologie des Skelettbaus von Pecora (Mammalia) und ihre Anwendung
zur Interpretation der Paläökologie des Neogens. *Paleontol. Evol.* (1990).
- 58. Sevim, A. and Y. Kiper, 1999 Yılı Çankırı Çorakyerler Kazısı, 22. Kazı Sonuçları Toplantısı, (ed
Basımevi, K.B.M.K.) 37-46 (Ankara, 2001).
- 59. Ünay, E., Bruijn, H. de & Suata-Alpaslan, F. Rodents from the upper Miocene hominoid locality
Çorakyerler (Anatolia). *Beitr. Paläont.* 30, 463–467 (2006).
- 60. Gaziry, A.W. Jungtertiäre Mastodonten aus Anatolien (Türkei). *Geolog. Jahrb. B.* 22, 1–143
(1976).
- 61. Mayda, S., Sevim Erol, A. & Yavuz, A. Y. Carnivora from Çankırı-Corakyerler Hominoid
Locality. 68th Geological Congress of Turkey, 06-10 Nisan/April 2015. 489 (2015).

- 62. Sevim Erol, A., Yavuz, A. Y. & Mayda, S. Çorakyerler Locality- The Center of the Youngest
Hominoids of Anatolia. in International Congress of Anthropological Sciences (ICAS), 9th-11th April
2015. Ankara, Turkey (2015).
- 63. Sevim Erol, A. et al. Çorakyerler Kazısı 2017 Yılı. 40. (ed Toplantısı, U. K. S.) 311-324 (2018).
- 64. Sevim Erol, A. et al. Çorakyerler Kazısı 2016 Yılı. 39. (ed Toplantısı, U. K. S.) 521-535 (2017).
- 65. Tarhan, E. et al. Zürih Üniversitesi Paleontoloji Müzesi'nde Bulunan Çorakyerler Suidae
Materyallerinin Revizyonu. MASROP E-Dergi 2018 12.2, 58-69 (2012).
- 66. Yavuz, A. Y. et al. Çorakyerler Lokalitesi Hystricidae Buluntuları. MASROP E-Dergi 2018
12.2, 2018: p. 70-75.
- 67. Kostopoulos, D. et al. Qurliqnoria (Bovidae, Mammalia) from the Upper Miocene of Çorakyerler
(Central Anatolia, Turkey) and its biogeographic implications. Palaeoworld.
doi.org/10.1016/j.palwor.2019.10.003 (2019).
- 68. Kostopoulos, D. et al. A new late Miocene bovid (Mammalia: Artiodactyla: Bovidae) from
Çorakyerler (Turkey). Fossil Record. doi.org/10.5194/fr-24-9-2021
- 69. Agustí, J. et al. A calibrated mammal scale for the Neogene of Western Europe. State of the art.
Earth-Sci. Rev. 52, 247-260 (2001).
- 70. Agustí, J. et al. "Late survival of dryopithecine hominoids in Southern Caucasus. J. Hum. Evol.
138, 102690 (2020)..
- 71. Suwa, G. et al. The first hominoid from the Maragheh Formation, Iran. Palaeobiodiv.
Palaeoenviron. 96, 373-381 (2016).
- 72. Emonet, E.G., Andossa, L., Taïso Mackaye, H., Brunet, M., Subocclusal dental morphology of
Sahelanthropus tchadensis and the evolution of teeth in hominins. Am. J. Phys. Anthropol. 153, 116-
123 (2014).

**Acknowledgments:** The second author acknowledges with much gratitude the researchers,
curators and collections managers who have provided access to fossils and comparative
collections in their

care since 1980. The authors thank Prof. Dr. Volkan Karabacak (Eskisehir Osman Gazi
University) for the DEM images in figure S9. **Funding:** The Çorakyerler excavations are
supported by the Turkish Ministry of Culture and Tourism, the General Directorate of
Cultural Heritage and Museums, Ankara University, and the Turkish Historical Society; we
are thankful to all of them. The second author acknowledges support from NSERC (grant
number RGPIN-2016-06761.) **Author contributions:** The first author is the PR of the
project, secured funding for excavations and lab analysis and directed excavations; The first
and second author are responsible for data curation, acquired funding and supervised
research; The second author conceptualized the analysis, conducted the investigation,
developed the methodology and prepared the original draft; All other authors contributed to
the preparation of the final manuscript. **Competing interests:** Authors declare no competing
interests. **Data and materials availability:** All data are available in the main text or the
supplementary materials. Contact the first author for access to fossil specimens. The new
taxon has the following Life Science Identifier:

<http://zoobank.org/References/> (to be determined)

**Supplementary Materials:**

Institutions and collections

Supplementary Text

Figures S 1 to S 10

Tables S 1 to S 7

References 40-70

Reviewers' comments:

Reviewer #1 (Remarks to the Author):

I appreciate the revisions to this manuscript. Generally speaking, it is improved from the previous version. There are still a few outstanding issues:

As requested, the authors now include a character matrix for their cladistic analysis. However, they still do not provide a table listing the hypodigms comprising each fossil OTU. Given the controversy surrounding the alpha taxonomy of most fossil species, it is important that readers know exactly which fossils were included in the analysis (e.g., listed by specimen number).

The authors also state that the character matrix is new and unique to this study, although they respond to Reviewer 2 that much of their work is based on the Nengo et al. (2017) and Pugh et al. (2022) matrices. The latter seems to be the case, and I would like to see citations for each character provided in the character matrix (or, at the very least, a list of citations in the caption for the table).

A more minor request for the table. As it currently stands, someone wanting to replicate this study would need to type up a few hundred character states from a pdf. Could the authors provide the matrix as a nexus file in the SOM? (this may just be an issue of how the reviewer package downloads, my apologies if it was actually provided as such).

Line 346-348: the authors state "The diversity of hominines in the eastern Mediterranean mirrors that among australopithecines in the Plio-Pleistocene hominin record in Africa."

-- I'm not sure why this is a relevant comparison to draw

Line 376-378: "Eastern Mediterranean hominines may represent a terminal radiation arising from one or more older hominines in Europe, analogous to the radiation of Paranthropus presumably from an Australopithecus-like ancestor."

-- Likewise, I don't see the need to draw parallels to hominins

Line 394-396: "Dental dimensions are standard mesiodistal and buccolingual measurements with canine and P3 measurements crown maximum and perpendicular dimensions. "

-- It seems like words are missing from this sentence?

Line 482: "Molds and casts were also made of this specimen"

-- To what end? Is this stated because they available for other researchers? were they used in the study somehow?

Line 518-519: "which are displaced but cannot be repaired on the original, repositioned and realigned"

--The last part of the sentence is hard to follow

Line 526: "Wewere able to virtually extract the canine using micro ct scans"

-- I assume the authors mean they were able to extract the canine using segmentation of micro ct scans. Please also reference software used for this segmentation (Avizo, VG Studio, etc)

Reviewer #2 (Remarks to the Author):

This MS is much improved from the original submission, and I appreciate the authors efforts to address many/most of my comments in this revision. The main descriptions and, in my view, the

argument that the fossils included here in the new genus *Anadoluivius* represent a distinct taxon closely related to *Ouranopithecus* and *Graecopithecus* seem very reasonable and they have been bolstered by the additional statistical analyses and photographs, as requested. However, there are still substantial shortcomings that need to be addressed, and additional clarification and details are needed throughout the supplementary materials and extended data. Therefore, this MS is still in need of another revision.

I again attach annotated PDFs with my comments throughout the main MS, the Supplementary Materials (SM), and the extended data (ED). I will summarize my main points/concerns here as well.

First, we will never see eye-to-eye on the cladistic analyses, and we will have to agree to disagree on some of this. However, the analyses being presented still have problems that need to be addressed before they are publishable. There are numerous issues here and they need to be addressed because they are then directly related to the discussion of evolutionary scenarios etc. The first is this: *Graecopithecus* is not included in the phylogenetic analyses within the main text, and its position is OUTSIDE of the hominines (and outside an *Ouranopithecus*/*Anadoluivius* clade) in the ED trees. Given that the authors are arguing for an Eastern Mediterranean clade, it is striking that it is not recovered in any of their trees. It makes some of the arguments in the Discussion regarding an Eastern Mediterranean clade a little more difficult and the authors can't make statements such as that on lines 291-292 that all analyses recover a hominine clade including all Eastern Mediterranean taxa...that statement is not true. *Graecopithecus* is never recovered with the other genera. So on the basis of their own trees, the authors need to be more cautious here. I think that they are probably correct that these taxa are closely related, but they can't claim relationships that their trees don't support without being more careful about their language throughout. In lines 243-244 and throughout the Discussion where they wish to suggest this clade, they need to cite another paper because these relationships aren't recovered here.

Second, the outgroups are described in a curious way on lines 441-442. Why are the outgroups "effectively" *Ekembo*, *Equatorius*, and *Nacholapithecus*? Why effectively? Either these taxa were assigned as successive outgroups (or not constrained successively) or they weren't. If all three of them were assigned as your Outgroup, then just state that this was the case and state whether or not relationships among them were constrained. If not, then this also needs to be stated clearly.

The biggest issue, of course, is still with the taxon sampling and the character ordering. Regarding taxon sampling, I recognize that you have to have cut-offs somewhere, and though I think the matrix should be expanded, I think the authors have made a reasonable attempt here and readers can judge the strength of their taxon sampling accordingly. What does need to be provided in the main text, however, is a clear and straightforward explanation around line 212 as to why these characters and taxa were chosen. You can provide additional details in the Methods (as you currently do), but we should have some brief statement as to why the analysis was run on these chars. and taxa within the main text.

As far as character ordering....I'm not sure if we are talking past each other or if there is a misunderstanding, but if I'm understanding the authors' arguments correctly (both in their text and their response to reviewers), they are essentially suggesting that ordering characters that are clearly along a transformation series is "introducing uncertainty and subjectivity in the form of a priori hypotheses of character state transformations, which ultimately affect the reproducibility of these analyses." Huh? How? There are 2 things here. First, either unordering OR ordering a character is providing an a priori hypothesis of how evolution is working for that particular feature. There is no less subjectivity by choosing to unorder the character, it is simply a different hypothesis. For some features, that makes sense. For instance, their character 4 Supraorbital margin....I don't quite understand what feature they are discussing (which points to another issue that I will come to), but as far as I can make out, there's not a clear transformation series by which a population would move from a sharp to a rounded and then to a rimmed supraorbital margin. Therefore, it is reasonable to

leave this as unordered, which essentially says that a population could go from having a sharp supraorbital margin to a rimmed one without ever passing through a rounded state. That seems like a reasonable model if there's no other information. On the other hand, take their character 19: Nasal bone length. This is a continuous character, whether the authors chose to measure it quantitatively or not. The nasal bones can be measured as the length from nasion to rhinion. Just because they chose to place the states in qualitative categories based on a more subjective assessment does not make the underlying feature any less continuous and it could easily be expressed more quantitatively. And any population with variation in nasal bone length is going to move through drift or selection towards either longer, shorter, or about the same nasal length from one generation to the next depending on the environmental pressures or chance events involved. Thus, a population would not evolve from having short nasal bones to having "extremely elongated" nasal bones without ever having moved through an intermediate "long" nasal bone state. So why would it make sense to model this character as unordered? Unless we are modeling Goldschmidtian hopeful monsters and discontinuous jumps in morphological features (in which case we can throw out the modern synthesis), choosing to unorder this character is a demonstrably poor way to model its evolution. It only makes sense to model it more accurately by making it ordered. In their response to reviewers, the authors seem to suggest that ordering requires a direction to be stated a priori....it does not. It only requires that a population pass through an intermediate step to get from one end of the transformation series to the other. The "direction" is determined by the outgroup(s), which set the initial polarity. So in the example above, if the Outgroup has "long" nasal bones, then it is only one step to move in EITHER direction to get to short or extremely elongated nasal bones. If the Outgroup has "short" nasal bones, then it is 2 steps to get to "extremely elongated." If the Outgroup has "extremely elongated" nasal bones, then it is 2 steps to get to "short" nasal bones. In contrast, all of these changes are only 1 step in an unordered analysis, and imply that there is no passing through intermediate states, which as I outline above, is in contrast to how evolution works on continuous characters as far as we understand since the Modern Synthesis. If I am misunderstanding their argument, then they need to articulate in much clearer terms why unordering is a better way to model evolution in these instances, but they cannot state that it involves additional subjectivity and a priori assumptions to order characters because that's simply not true. Both ordering and unordering involve assumptions and are an attempt to model evolution...one does not make fewer assumptions than the other, the assumptions are simply different and one or the other can be a much better model depending on the character. At the end of the day, it would be a stronger argument to just not get into it in this MS and simply state in lines 451-452 that they ran an analysis with all characters unordered and an analysis with all characters that could be reasonably modeled as a character state series as ordered and then just leave it at that. Along those lines, there are many more than only 8 characters that should be ordered. If the authors are going to run an ordered analysis, it needs to be done with ALL characters that could be reasonably ordered, not a selected/cherry-picked small number of 8 characters. It shouldn't be up to the reader to have to do their job for them....the strongest argument the authors can make is to run the analysis both ways to their fullest extent. So if they are going to run the ordered analysis (which is necessary in my view), they need to order as many chars. as is reasonable and let any interested reader adjust or make the special pleading to unorder this character or that later. It results in a stronger argument to take the thought experiment to its fullest extent rather than engaging in special pleading from the beginning, which is what seems to be happening here.

I have highlighted a number of characters in Table S4 and Table S5 that I think should be ordered. I have also indicated the way that they characters should be coded to represent the transformation series. While there might be a couple that could reasonably be argued to stay unordered, the vast majority of these should be ordered in the next revision unless the authors can provide reasons otherwise using arguments about why a population would not move through the intermediate state that is being modeled. For example, the authors claim a few "counter-intuitive" characters in lines 455-459, but never explain why it is unreasonable to assume that a population would NOT pass through an intermediate stage of enamel thickness when evolving from thin to thick enamel, for example. Why is it unreasonable to assume that a population would move from having no ear tube to a complete ear tube without ever having an intermediate morphology? The answer in the response to

reviewers seems to suggest that because they have clades with both thin and thick enamel, for instance, and yet no intermediate enamel, that means there must not have been an intermediate state within that clade. This is like saying because there are small and very large apes in a given clade, but no apes of in-between size, that there was never an ape population that was intermediate in size. That, of course, makes very little sense and is only a consequence of sampling. Just because we don't have the intermediate condition preserved does not mean that the intermediate character state was never passed through. So this argument does not pass muster, in my opinion.

There is also a lack of clarity as to the characters and the character state definitions. I asked how many of these were quantitatively assessed in my last review, and I asked how polymorphisms were handled etc. In the response to reviewers, the authors say that all the characters are qualitative and claim that Nengo et al. used only qualitative characters as well, but I went back and re-read that paper and that is not true. There are an admittedly small number of quantitatively coded chars., but they are there. In addition, they provide clear evidence that a number of other characters were quantitatively assessed, with quantitative cut-offs for their qualitative chars. in a number of places...for instance, ulnar olecranon process size as a % of sigmoid notch: very short <50%, short (51-80%), long (80-100%, and very long (>100%), and they point to a definition in Rossie and MacLatchy. Or anterior calcaneal length as >1/3 of total length, ~1/3 of total length, or < 1/3 total length. The authors should do this for their characters here as much as possible to increase repeatability. Many of the characters and character state definitions here list states such as "short" or "wide" but provide no details as to how those cut-offs were determined. In some cases, it might not be possible or necessary, but more detail should be provided as much as possible. Again, using the example of nasal length above, even if coded qualitatively, I assume this character essentially an assessment of nasal bone length (nasion-rhinion)? This could be made clearer and should be stated. And is short just using the eyeball-ometer, or is there a quantitatively measured relative nasal bone distance that is considered short? Is short ~0-50% of the total face length (nasion-prosthion) or how was this assessed? Right now, it is entirely subjective and I have little confidence that anyone other than the authors could sit down and code a new specimen accurately using these characters because they are not clearly defined.

My other great concern with the ordered character analysis is the character state transformation series. In the 8 characters that are stated to be ordered (Chars. 7, 10, 22, 37, 38, 41, 43, and 106), the states listed are not in the order of the assumed transformation series. So character 10, interorbital space (which I assume is interorbital breadth...this should be better described) has 3 states listed as 0=broad, 1=narrow, and 2=intermediate. This is fine for the unordered analysis, because the analysis does not assume any transformation series and allows going from broad to narrow in a single step. However, when running an ordered analysis, the series must be defined as 0=broad, 1=intermediate, and 2=narrow OR 0=narrow, 1=intermediate, and 2=broad. I have no idea if this was done because it is not stated anywhere. If left in the same original order as listed in Table S5, then this states that a population must pass through a narrow character state to get from broad to intermediate. This would not make any sense. So the authors should re-write their character and character state definitions to make any ordered characters clear OR they should provide separate tables listing their character states for the ordered and unordered analyses. They should also provide matrices for each of their analyses somewhere and provide these as nexus files, not as a text table as in Table S6. Right now looking at Table S6, if the char. States were left in the same order as listed in Table S5, then that is a serious problem for the ordered analysis. For example, Pongo has state 2 listed for Character 10 in the text file given as Table S6. So does this mean that Pongo was coded as having an "intermediate" char. State for Character 10 (interorbital breadth)? If so, then shouldn't this be a "1" for the ordered analysis, in between "broad" and "narrow"? This needs to be clarified throughout. The easiest would be to provide multiple tables or to somewhere provide alternate char. States for the ordered analysis. And again, for transparency, provide separate matrices demonstrating the coding for both the ordered and unordered runs.

Lines 455-470 state that multiple runs were performed adding additional taxa, but they never clearly

state which analysis is which. Be specific and state clearly what is going on in each run. Similarly, the captions in ED Figure 6 need some additional details so that the reader knows what the analysis is and what the tree represents (which taxa were added, ordered/unordered, etc.). Finally, ED figure 5 only lists synapomorphies for the unordered analysis. After the ordered analysis is re-run taking into account the addition of the ordered chars. as suggested above, the synapomorphies reconstructed by the ordered analysis should be provided here as well.

The phylogenetic analysis in this paper is crucial because much of the Discussion relies on it, and that is why it must be updated and be made more transparent and repeatable.

Speaking of the Discussion, my last major point is this: it is fine for the authors to discuss the Mediterranean taxa and dryopithecines as hominines, as their cladograms (and some others) support this/these hypotheses as well (although see my point about *Graecopithecus* above). I also think that the biogeography discussion, even if I disagree, is a reasonable point to make. However, as much as the authors discount *Nakalipithecus* by suggesting it is distinct, that's not really a very good argument. Of course it is distinct from the Mediterranean taxa....that is why it is in a separate genus. If you want to make the case that it has nothing to do with it, some sort of quantitative assessment should be provided. The trees here are one piece of evidence, and maybe should be referenced. The differential diagnosis helps demonstrate that *Nakalipithecus* is a distinct genus from the eastern Mediterranean genera, which again, I don't think is in dispute. It would be a much stronger argument that *Nakalipithecus* has no relationship to *Ourano* and *Anadoluvius* if it was demonstrated more quantitatively/objectively using some of the features you analyze here. I note that *Nakalipithecus* is not in the quantitative analyses of dental comparisons etc. in the SM, while other Eurasian ape are. Why not? Similar comments for *Chororapithecus* and *Samburupithecus*, which I think are less likely to have anything to do with the Eastern Mediterranean taxa, but should probably still be included in the analyses, nonetheless.

The part of Conclusion that should be modified, however, is in lines 371-375. There is no reason to even bring up the suggestion of a connection with the earliest African hominins. None of the trees suggest that these apes are anything other than stem hominines. The noted similarities in canine reduction (and enamel thickness) are clearly independent evolutionary parallelisms, and canine reduction is not happening in the same way in the Eastern Mediterranean apes and the earliest hominins, so this is all bit misleading. To use a different paleoprimates example, this is like saying that both plesiadapiforms and omomyoids show evidence of incisor enlargement and implying that means something phylogenetically. The details of the incisor enlargement and the loss of other teeth is totally different in those 2 groups and has very little phylogenetic valence. Similarly, the details of the canine reduction here are clearly different or occurring in parallel (at best) with early hominins. They are on shaky ground suggesting any connection with hominins when their own cladograms suggest nothing of the sort. Lines 373-375 should be deleted.

More minor comments (see also my attached annotated PDFs):

-Throughout the main text (and in the SM and ED), additional details and clarifications are needed. For instance, page 9 of the main text talks about "...lower canine size to m2 size...". Ok...but how is lower canine size and m2 size being defined? Lower canine to m2 size meaning what ratio? Is this lower canine area (=L x W) divided by m2 area? Be specific please. I have noted numerous parts throughout all of the documents where additional clarification is needed regarding measurements and additional specificity is needed to make clear exactly what measurements or taxa are being compared.

-Some of the figure references seem off throughout the main text, SM, and ED. I'm sure this is because figures have changed and moved around during the revision process....just please double-check and make sure they all line up correctly.

-Lines 280-281 state that this is the "most comprehensive" analysis of eastern Mediterranean taxa. Comprehensive in what sense? Certainly not in the total number of characters or total number of taxa included. This statement needs to be qualified.

-Reference 9 should be updated as the recent JHE paper, not an abstract.

-Similarly, there is an un-numbered figure near the beginning of the SM showing the cranium in situ. Should this be Fig. S1? If so, then all other figures need renumbering. I offer a possible suggestion in the annotated PDF, so maybe this needs to be moved or integrated with the current S4?

-In the SM, the dentition of CO 2100/2800 should be described in much more detail and compared with CO 205 and other specimens.

-Fig. S2- Are these both females? Please state the sex of each specimen in the caption so we know we are comparing apples to apples in terms of sexual dimorphism.

-Fig. S5- Are these all supposed to be males? If so, please state that in the figure caption. Otherwise, I think the female palate from the cranium 2100/2800 should be included in this comparison.

-SM lines 358-359- "more consistent morphologically with *Graecopithecus*" Are there other upper premolars and molars of *Graecopithecus* to compare it to? How would we know?

-ED throughout- there needs to be a table where all of your measurements are clearly defined. A number are defined in the Methods, others in figure captions, and they are not easy to find. If you could just compile them all into one table, that would be extremely helpful and allow you to reference that table in all of the captions where you provide measurements (which are very welcome, by the way)

-ED Table 3- Thank you for providing the ANOVA comparisons. Some summary statistics would also be helpful somewhere so that the reader can quickly and easily judge what direction the significant directions are in. Please provide the means and ranges for the taxa included.

-ED Table 4- Again, great that you are providing the raw measurement values here. However, as stated, these are not the values that are being put into the PCA...the relative values are instead. Can you provide a table with the relative values as well? Also, I assume the GM was calculated from all 12 raw values provided in this table? Please be specific and clear about what measurements and values went into the GM.

-ED Figure 2- Please be more specific in the caption as far as what is going on for each axis. e.g., something like "PC 1 is driven primarily by mandibular arch breadth such that broader mandibles are at the positive/negative end of PC1 and narrower mandibles are at the positive/negative end of PC1; PC2 is driven largely by symphyseal M1-M2 distance such that a longer symphysis is at the positive/negative end of PC2 and a shorter symphysis is at the negative/positive end of PC2."

-ED Figs 3-4- Please be more specific as to what the indices actually are. Is canine/m2 size actually canine area/m2 area? What is the ratio here? And if it is canine breadth/m2 area, then the square root of m2 area should be used to keep everything in the same units. In any case, measurements and ratios must be more clearly defined.

We respond to the specific points of the reviewers in detail in the annotated pdfs and in annotations in the text of the reviews. We summarize these responses/rebuttals here. The reviewer comments are in red.

Reviews:

However, they still do not provide a table listing the hypodigms comprising each fossil OTU.

We state that all available specimens in every taxon were analyzed. This includes hundreds of specimens. We will provide a table with more details of locality and repository for each taxon.

The authors also state that the character matrix is new and unique to this study, although they respond to Reviewer 2 that much of their work is based on the Nengo et al. (2017) and Pugh et al. (2022) matrices. The latter seems to be the case, and I would like to see citations for each character provided in the character matrix (or, at the very least, a list of citations in the caption for the table).

This is a misinterpretation of our text. We are clear that all data are original to this analysis but that the methods used are consistent with many recent analyses. This is in contrast to the reviewer's opinion that our analysis does not meet current standards in the field. However, we recognize that we could add additional tests such as bootstrapping and explanations of analytical specifics such as the use of Wagner trees, which allows for reversals, etc.

Could the authors provide the matrix as a nexus file in the SOM?

Done. This was in all the versions. We do not know why the reviewer was unable to find it.

Line 346-348: the authors state "The diversity of hominines in the eastern Mediterranean mirrors that among australopithecines in the Plio-Pleistocene hominin record in Africa." -- I'm not sure why this is a relevant comparison to draw

We think that readers would find it interesting that there are repetitive patterns in hominid evolution. This is akin to references to parallel evolution of similar ecomorphs on different continents, etc. There is also the implication that precedent lend some credibility to our interpretation. We are not saying that the pattern similarity proves anything, but that the pattern we are describing is known to occur in closely related lineages. We find this to be an interesting observation.

And any population with variation in nasal bone length is going to move through drift or selection towards either longer, shorter, or about the same nasal length from one generation to the next depending on the environmental pressures or chance events involved. Thus, a population would not evolve from having short nasal bones to having "extremely elongated" nasal bones without ever having moved through an intermediate "long" nasal bone state. So why would it make sense to model this character as unordered? Unless we are modeling Goldschmidtian hopeful monsters and discontinuous jumps in morphological features (in which case we can throw out the modern synthesis), choosing to unordered this character is a demonstrably poor way to model its evolution. It only makes sense to model it more accurately by making it ordered. In their response to reviewers, the authors seem to suggest that ordering requires a direction to be stated a priori...it does not. It only requires that a population pass through an intermediate step to get from one end of the transformation series to the other. The "direction" is determined by the outgroup(s), which set the initial polarity. So in the example above, if the Outgroup has "long" nasal bones, then it is only one step to move in EITHER direction to get to short or extremely elongated nasal bones. If the Outgroup has "short" nasal bones, then it is 2 steps to get to "extremely elongated." If the Outgroup has "extremely elongated" nasal bones, then it is 2 steps to get to "short" nasal bones. In contrast, all of these changes are only 1 step in an unordered analysis, and imply that there is no passing through intermediate states, which as I outline above, is in contrast to how evolution works on continuous characters as far as we understand since the Modern Synthesis. If I am misunderstanding their argument, then they need to articulate in much clearer terms why unordering is a better way to model evolution in these instances, but they cannot state that it involves additional subjectivity and a priori assumptions to order characters because that's simply not true. Both ordering and unordering involve assumptions and are an attempt to model evolution...one does not make fewer assumptions than the other, the assumptions are simply different and one or the other can be a much better model depending on the character. At the end of the day, it would be a stronger argument to just not get into it in this MS and simply state in lines 451-452 that they ran an analysis with all characters unordered and an analysis with all characters that could be reasonably modeled as a character state series as ordered and then just leave it at that. Along those lines, there are many more than only 8 characters that should be ordered. If the authors are going to run an ordered analysis, it needs to be done with ALL characters that could be reasonably ordered, not a selected/cherry-picked small number of 8 characters. It shouldn't be up to the reader to have to do their job for them....the strongest argument the authors can make is to run the analysis both ways to their fullest extent. So if they

are going to run the ordered analysis (which is necessary in my view), they need to order as many chars. as is reasonable and let any interested reader adjust or make the special pleading to unorder this character or that later. It results in a stronger argument to take the thought experiment to its fullest extent rather than engaging in special pleading from the beginning, which is what seems to be happening here. I have highlighted a number of characters in Table S4 and Table S5 that I think should be ordered. I have also indicated the way that they characters should be coded to represent the transformation series. While there might be a couple that could reasonably be argued to stay unordered, the vast majority of these should be ordered in the next revision unless the authors can provide reasons otherwise using arguments about why a population would not move through the intermediate state that is being modeled. For example, the authors claim a few "counter-intuitive" characters in lines 455-459, but never explain why it is unreasonable to assume that a population would NOT pass through an intermediate stage of enamel thickness when evolving from thin to thick enamel, for example. Why is it unreasonable to assume that a population would move from having no ear tube to a complete ear tube without ever having an intermediate morphology? The answer in the response to reviewers seems to suggest that because they have clades with both thin and thick enamel, for instance, and yet no intermediate enamel, that means there must not have been an intermediate state within that clade. This is like saying because there are small and very large apes in a given clade, but no apes of in-between size, that there was never an ape population that was intermediate in size. That, of course, makes very little sense and is only a consequence of sampling. Just because we don't have the intermediate condition preserved does not mean that the intermediate character state was never passed through. So this argument does not pass muster, in my opinion.

This is probably the main objection from this reviewer. These are our responses to the points made by this reviewer.

While this is perfectly sensible evolutionary theory, unfortunately the fossil record is not one of successive populations but is wildly incomplete. In the hundreds of thousands to millions of years between samples, multiple intermediates and reversals would have evolved. Moving from short to long to intermediate by allowing the algorithm to find the shortest path is easily accommodated within evolutionary theory simply by recognizing that the intermediate state between long and short is not preserved, which we know is the case for much of the fossil record. Perfectly intuitive Intermediates are rare in the fossil record.

Many of our characters have as the primitive condition an intermediate character state. To order these characters requires that, in some circumstances depending on the character state of successive clades, that the transformation series move from intermediate to one extreme then to the other. Our "all ordered" analyses are always less parsimonious because of this. In the case where the primitive condition, as defined by the outgroups, as intermediate, it is reasonable to leave these characters unordered.

Fair enough. What we should have said it that no assumption of a transformation series is justified a priori for fossil samples separated by huge spans of time. While premaxillary length probably did experience transformational sequence passing through intermediate steps from short to long, from intermediate to long or from long to very long, all the steps are not preserved in the fossil record. And when reversals occur, and they do with considerable frequency in the hominoid fossil record, intermediate steps are rarely documented in the fossil record.

We have now done exactly what the reviewer proposes and run the analysis fully with all characters ordered, unordered and ordered as per the reviewer's recommendations, with the caveat expressed above.

We have recoded all the characters to reflect an intuitive transformation series (small to large and vice versa, etc.).

To repeat from earlier, fossil samples are not successive populations. Most intermediate steps between fossil samples are unknown, they are separated by enough time to allow for multiple reversions and convergences.

We know that the intermediate condition is the condition in the outgroup at least 20 times for multistate characters. So, it must be the case that intermediates, if they existed between two extremes, are missing. If they were preserved, they would by definition be convergences and thus not homologous with the primitive intermediate condition. It does make sense to order characters based on hypothetical transformation series that are not observed in the data. Where there is evidence of the potential for an ordered transformation series without violating the assumptions of outgroup polarity determination, we ordered those characters. Stated another way, without direct evidence of a specific transformation series beyond the intuition that there must always be intermediates, there is no way to determine if an intermediate condition observed in one taxon is homologous with the hypothetical intermediate condition that might have been present between two extremes, but it not preserved in the fossil record.

We have added text in the methods that responds to this reviewers' comments. We believe that much of this debate reflects difference of philosophy in cladistic analysis. All programs allow for the option of both ordered and unordered characters because both options are considered appropriate. Philosophical differences sometimes lead those with opposing views to be dogmatic, but they should not lead to one side preventing publication by the other side.

From the revised Methods:

[revised manuscript text omitted]

Other "counter-intuitive" character state transformation series include the development of the ectotympanic tube (absent, partial or complete), enamel thickness (thin, intermediate, thick), and lumbar vertebrae number (seven to six to five to four to three), none of which are supported by the majority of most parsimonious analyses of hominoids in numerous publications. Must these transformations have had intermediates that are simply missing from the fossil record, or it is possible to go from, for example, seven to four to five lumbar vertebrae by simple conversion of vertebral type (e.g. thoracic to lumbar). Humans have a median number of five lumbar vertebrae but four is not uncommon. Great apes have a median number of three but four is common. Given this variability a "jump" from three to five or vice versa is certainly feasible. We do not understand the genetics of these character states sufficiently to know how transformation occurs and if "jumps", which are more a function of how we define character states than anything else, are likely, probable or impossible. Finally, it could be argued that allowing a character state to "jump" to a subsequent state without an intermediate condition based purely on parsimony is akin to rejecting Darwinian gradualism in favor of Goldschmidian "hopeful monsters". However, Goldschmidt was referring to species origins and not individual character states. Hypothesizing character state transformations that do not pass through intermediate states does not violate the basic principles of Darwinian evolution.

There is also a lack of clarity as to the characters and the character state definitions. I asked how many of these were quantitatively assessed in my last review, and I asked how polymorphisms were handled etc. In the response to reviewers, the authors say that all the characters are qualitative and claim that Nengo et al. used only qualitative characters as well, but I went back and re-read that paper and that is not true. There are an admittedly small number of quantitatively coded chars., but they are there. In addition, they provide clear evidence that a number of other characters were quantitatively assessed, with quantitative cut-offs for their qualitative chars. in a number of places...for instance, ulnar olecranon process size as a % of sigmoid notch: very short <50%, short (51-80%),

long (80-100%, and very long (>100%)), and they point to a definition in Rossie and MacLatchy. Or anterior calcaneal length as >1/3 of total length, ~1/3 of total length, or < 1/3 total length. The authors should do this for their characters here as much as possible to increase repeatability. Many of the characters and character state definitions here list states such as "short" or "wide" but provide no details as to how those cut-offs were determined. In some cases, it might not be possible or necessary, but more detail should be provided as much as possible. Again, using the example of the documents where additional clarification is needed regarding measurements and additional specificity is needed to make clear exactly what measurements or taxa are being compared.

We have added details for every character to make it easy for readers to replicate and assess our observations. We also state that characters with a size attribute (large, small, etc.) are assessed relative to adjacent structures. We do not have quantitative data on all of these, but even if we did, defining characters quantitatively is not without other problems. We note in the methods that Pugh (2022) finds problems with discretized character defined quantitatively and uses continuous characters scaled to molar length. This is also problematic given allometric concerns. Simple ratios fail to account for allometry and issues such as differential dental size among taxa (megadontia vs microdontia), which are prevalent in this data set.

Some of the figure references seem off throughout the main text, SM, and ED. I'm sure this is because figures have changed and moved around during the revision process...just please double-check and make sure they all line up correctly.

There has been a fair amount of reshuffling of figures. We trust that they are all legible currently. Once we have settled on an outcome for this ms, we will adjust the figures accordingly.

-Lines 280-281 state that this is the "most comprehensive" analysis of eastern Mediterranean taxa. Comprehensive in what sense? Certainly not in the total number of characters or total number of taxa included. This statement needs to be qualified.

Deleted

-Reference 9 should be updated as the recent JHE paper, not an abstract.

Done

-Similarly, there is an un-numbered figure near the beginning of the SM showing the cranium in situ. Should this be Fig. S1? If so, then all other figures need renumbering. I offer a possible suggestion in the annotated PDF, so maybe this needs to be moved or integrated with the current S4?

This was included in error. Part of this figure, showing the specimen in situ, is reproduced in figure S3.

-In the SM, the dentition of CO 2100/2800 should be described in much more detail and compared with CO 205 and other specimens.

Done

-Fig. S2- Are these both females? Please state the sex of each specimen in the caption so we know we are comparing apples to apples in terms of sexual dimorphism.

Done

-Fig. S5- Are these all supposed to be males? If so, please state that in the figure caption. Otherwise, I think the female palate from the cranium 2100/2800 should be included in this comparison.

Done

-SM lines 358-359- "more consistent morphologically with Graecopithecus" Are there other upper premolars and molars of Graecopithecus to compare it to? How would we know?

The three citations following this statement cover this issue in detail. We do not feel that needs to be repeated here.

-ED throughout- there needs to be a table where all of your measurements are clearly defined. A number are defined in the Methods, others in figure captions, and they are not easy to find. If you could just compile them all into one table, that would be extremely helpful and allow you to reference that table in all of the captions where you provide measurements (which are very welcome, by the way)

We think all the measurements are defined in the legend to extended table 5a.

-ED Table 3- Thank you for providing the ANOVA comparisons. Some summary statistics would also be helpful somewhere so that the reader can quickly and easily judge what direction the significant directions are in. Please provide the means and ranges for the taxa included.

Done

-ED Table 4- Again, great that you are providing the raw measurement values here. However, as stated, these are not the values that are being put into the PCA....the relative values are instead. Can you provide a table with the relative values as well? Also, I assume the GM was calculated from all 12 raw values provided in this table? Please be specific and clear about what measurements and values went into the GM.

Very frequently only the relative values are provided, making it impossible to calculate the raw values. We think that readers can easily calculate the relative values, and even choose their own version of a GM, from these raw values.

-ED Figure 2- Please be more specific in the caption as far as what is going on for each axis. e.g., something like "PC 1 is driven primarily by mandibular arch breadth such that broader mandibles are at the positive/negative end of PC1 and narrower mandibles are at the positive/negative end of PC1; PC2 is driven largely by symphyseal M1-M2 distance such that a longer symphysis is at the positive/negative end of PC2 and a shorter symphysis is at the negative/positive end of PC2."

Done

-ED Figs 3-4- Please be more specific as to what the indices actually are. Is canine/m2 size actually canine area/m2 area? What is the ratio here? And if it is canine breadth/m2 area, then the square root of m2 area should be used to keep everything in the same units. In any case, measurements and ratios must be more clearly defined.

In the legend to extended data figure 3 size is defined as $\ln \times \text{bd}$. We added a clarification that all ratios are scaled to the same power.

Main text annotated pdf:

line 101: I think you are referring to Fig. S5. Please double check your figure references.

Corrected

line 146: lower canine to m2 size meaning what ratio? Is this lower canine area (=L x W) divided by m2 area? Be specific please. We explain this measurement and ratio in the legend.

line 148: the individual geometric means made up of how many measurements and in what part of the anatomy? A couple of quick additional details are needed here, not just in the Methods or Extended data. The geometric mean is explained in the text and legends.

line 158: outside the quartile of Ouranopithecus? Be specific here....

Corrected

line 204: Given all of the comparisons above and discussion below, why isn't Graecopithecus included in these analyses? added, see responses and revised text

line 212: How or why were these characters and taxa (in particular) chosen? Provide at least some sort of justification for compiling the matrix this way.

Done (see text).

line 222: This means very little. Of course there are fewer steps when you don't have to go through intermediate states to go from small to large. And since the indices are in part dependent on the number of steps, it is also not an apples to apples comparison. In fact, tree statistics are largely unhelpful and highly dependent on the number of characters, character states, and taxa within the trees such that there is an inverse relationship: the higher the number of taxa, characters, and character states, the lower the CI values, and there is no proven correlation with accuracy either. I would simply delete this statement....it is misleading.

Deleted.

line 226: Ok, on what basis? At some point, there needs to be a discussion about this because it is very important for your biogeographic argument.

The cladistic analysis has been reworked, largely following the reviewer's suggestions. The statement about *Nakalipithecus* follows from the results of the cladistic analysis.

line 231: Figure S5 is currently a comparison of palates. Please revise....i don't see any figure showing the change in char. states across the try.

Fixed

line 243: sister taxa implies two taxa to me. Maybe just say a distinct clade instead. And since *Graecopithecus* isn't included in your preferred phylogenetic analysis, and isn't recovered in a clade with *Ouranopithecus* and *Anadoluvius* in the expanded analysis, you should cite another paper to support this idea.

Cited

Comprehensive in what sense? Certainly not total number of characters or taxa included. This statement needs to be qualified. Deleted.

line 292: Not true. *Graecopithecus* is outside the recovered hominine clade in ED fig. 6.

Corrected in the revised analysis

line 305: Ok, but how come *Nakalipithecus* isn't included in any of the metric analyses in this paper? If you want to make this case, some sort of quantitative assessment should be provided. The differential diagnosis helps demonstrate that *Nakalipithecus* is a distinct genus from the eastern Mediterranean genera, which I don't think is in dispute. This paper might not be the place for it, but it is not a strong argument that *Nakalipithecus* has no relationship to *Ouro* and *Anadoluvius* until you can demonstrate this more quantitatively/objectively.

Nakalipithecus is not well enough preserved to be included in the quantitative analyses. It lacks too many of the measurable dimensions to generate a meaningful geometric mean. All the analyses would have to be redone as simple ratios to include *Nakalipithecus*, and this is unacceptable because simple ratios fail to account for allometry and co-variation. Besides that, *Nakalipithecus* is in all the cladograms and always falls outside the Hominidae. In so far as a numerical cladistic analysis is objective, this relationship has been demonstrated here.

line 309: Similar comment as above. I agree, but it should be demonstrated in more quantitative detail in the future.

Same problem as with *Nakalipithecus*, but even worse. The goal here is to figure out the relations of *Anadoluvius* in comparison with all relevant taxa, not to work out all late Miocene hominid relationships. That would require separate analyses designed to accommodate these more poorly preserved samples, which is certainly doable but beyond the scope of this research. We note as well that the original authors of those two taxa have not tried a phylogenetic analysis nor a detailed quantitative analysis, presumably recognizing as we do that they are currently not well enough known (Pugh does include *Nakalipithecus* in their analysis but does not provide any quantitative analysis of this taxon).

line 373: OK, but canine reduction is not happening in the same way. This is a bit misleading. To use a different paleoprimates example, this is like saying that both plesiadapiforms and omomyoids show evidence of incisor enlargement and implying that means something phylogenetically. The details of the incisor enlargement and the loss of other teeth is totally different in those 2 groups and has very little phylogenetic valence. Similarly, the details of the canine reduction here are clearly different or occurring in parallel (at best) with early hominins. The authors should be careful here and rephrase to make clear that this is not a synapomorphy with early hominins. Better yet, they should just delete the statement "recalls the earliest hominins". Similarly, the following statement is highly controversial and speculative, even going by their own cladograms which the authors seem to put a lot of stock in. I think they have reasonable grounds for suggesting their preferred scenario of dryopithecines including an Eastern Mediterranean clade are stem hominins. They are on much shakier ground suggesting any connection with hominins when their own cladograms suggest nothing of the sort. These sorts of speculations should be deleted.

As noted in our reply to the review provided by this reviewer, there is no basis for concluding that canine reduction is homoplasious in hominins and Balkan hominins. This is pure conjecture. We never state that there is conclusive evidence that Balkan hominins are related to hominins. We simply state, and we believe it would be inappropriate to ignore, that these similarities exist but their significance is unclear and will continue to be pending further discoveries. It is surprising to us that this reviewer objects to our suggestion that there might be some information within the sample of Balkan hominins relevant to hominin origins. As we noted, if these samples had been recovered from Kenya or Ethiopia few would doubt their relevance to hominin evolution.

line 400: where on the symphysis? Can you name an osteological point?

We describe the precise location of this spot. We don't know if there is an osteometric point for this spot.

line 400: So how many measurements in total? Be specific! I assume all 12 that are listed in the table, correct?

We specify that all 12 were used.

line 441: Why effectively? If all three of them were assigned as your Outgroup, then just state that this was the case. If not, then this also needs to be stated clearly.

We have forced Ekembo, Equatorius and Nacholapithecus into a clade in Mesquite (see cladograms) and compared this configuration with that obtained from TNT (we could not define multiple outgroups in TNT). The tree statistics were the same as with the single outgroup.

line 451: This makes no sense. There is no less uncertainty or subjectivity or a priori assumptions being made by unordering characters....you are simply making different assumptions that are, in my view and that of many others, totally unreasonable in modeling how evolution works. Your argument seems to be "Well, since I haven't found a fossil with an intermediate maxillary length, that must mean that no population went through an intermediate state to get from having a short to having a long maxilla." Unless we are resurrecting Godlschmidian hopeful monsters, that is an unreasonable assumption. Take Gigantopithecus, a very, very large ape. It is a close relative of Sivapithecus and Pongo, both large apes. Should I assume that a population of pongines went from Sivapithecus/Pongo size to Gigantopithecus size in one step without ever passing through an intermediate size range just because we don't have the fossils to demonstrate this intermediate size range? That is what you are arguing and I find it totally unreasonable and a poor way to model evolution by drift or natural selection. For quantitative traits, selection (or drift) is going to drive populations in one direction or another through intermediate stages over time/generations, not through large discontinuous jumps....whether or not those stages are preserved in the fossil record is irrelevant. They surely existed unless you believe evolution works in giant discontinuous leaps, in which case do not pass go, do not collect \$200. I would delete this whole justification (it is unconvincing) and simply state that you ran it both ordered and unordered.

This section has been thoroughly revised. Please see our reply in the review and our revised text. In particular, see our revised phylogenetic analysis section in the methods. We are taken aback by the tone of the reviewer's comments and their reference to a board game. We are trying as hard as we can to take all the reviews seriously but when there is so much emotion it is hard to avoid the thought that there may be more to the reviewers' objections than simply what they have read in our text.

line 454: What fossil evidence demonstrates that this is "not the case"? This is just arm-waving. If the argument is that because you have a clade containing taxa with only absent or complete ear tubes, or only thin or thick enamel, that the intermediate states between these end points never existed then that is a ridiculous assumption. It is like saying that I have a clade of monkeys that contain two states: red tails and blue tails. Therefore, there was never any intermediate state between these two fixed states. In reality, a population of monkeys with red tails evolved into a population of monkeys with blue tails by passing through a polymorphic condition where the population had both red-tailed and blue-tailed individuals. Again, the only way you don't pass through this intermediate state is by imagining a discontinuous jump in one generation. If that is what you are arguing, then the modern evolutionary synthesis is being thrown out the window. That is why ordering characters when there is a clear or reasonable inference of directionality is a much better way of modeling the characters....it is in line with the modern synthesis. Unordered characters are not.

we have deleted the reference to uncertainty and subjectivity in ordered analyses, and we now include them. But the hyperbole here is again disconcerting. As noted, the fossil record does not record most intermediates, the span of time between samples is enormous and there is more than adequate time for similarities to arise either by shared descent or homoplasy.

line 455: What ape is being coded as having anything other than a complete ear tube?

This is an example of the lack of intermediates in the fossil record. It is a good example of how a character could "jump" an apparent intermediate condition. We know nothing about the genetics or developmental biology of ectotympanic tubes. Just because there are catarrhines with incompletely ossified ectotympanic tubes does not mean that the transformation from absent to fully ossified tubes must have gone through the condition of partially ossified. The tube is present in cartilaginous form, which does not preserve in fossils. A simple mutation might well result in the complete ossification of the tube that was cartilaginous in the ancestor, thus jumping from absent to present, without the "intermediate" condition. Again, the important point is that the presence of a condition that we perceive as intermediate in a fossil taxon is not proof that all taxa must have gone through that condition in the transformation from one extreme to another. It is somewhat naïve to think that we know enough about the genetics and development of attributes to know exactly what the transformation series must have been.

line 457: ??? Again, this is irrelevant. Just because a tree reconstructs a clade with both thin and thick enamel doesn't mean there wasn't an intermediate state at some point in time. It only means that intermediate population wasn't preserved in the fossil record. Does the reviewer know enough about the genetics of dental development to be able to say with certainty that the change from thin to thick enamel must have passed through an intermediate stage? Between *Ardipithecus* and *Australopithecus* we go from thin to hyperthick. Is it not possible that a single or a few genetic changes could prolong amelogenesis a few days and produce a difference that is quantitatively the difference between what is coded as thin vs thick? In this case there are only intermediates because there are other taxa that have intermediate enamel thickness, not because whatever came between *Ardipithecus* and *Australopithecus* must have had thin enamel. Do we know if the position of the incisors relative to the nasal aperture must transform from within the

margins to inline to outside the margins, or the other way around? Why is it impossible to imagine that a small change in timing and duration or premaxillary development could result in a change from outside to inside without an intermediate?

line 462: Why only 8? If you are going to run an ordered analysis, at least do it right. You can't do a half-baked one. See my comments on Tables S4 and S5. A whole bunch of other characters should be ordered. Either run the ordered analysis to its fullest reasonable extent or do not. But you can't reasonably argue that nasal length, for instance, goes from short to long without ever being intermediate in length within a population. And it is unreasonable for you to say that "well, the reader can just run the matrix how they want to and it's not our job to do it." That is wrong and the reader shouldn't have to do your job for you. You should run the ordered analysis in full and then if readers want to disagree or change some coding, then that's fine. But the strongest case is to say that you ran both the unordered and ordered analyses to their fullest extent and then let any interested reader decide what to cherry-pick or take out. If you are cherry-picking and engaging in special pleading from the beginning, as you are here, then you are presenting a poor argument.

We object to the claim that we are cherry-picking, as we have what we feel are sound and justified reasons for our decisions. Cherry-picking implies that we are choosing characters and approaches that ensure the outcome we desire, which is unconscionably bad science. Is that the accusation here? Half-baked? Really? At any rate, we have now followed the reviewers' suggestions in terms of ordering as far as we can given the emergent polarities of various characters and following the logic of the reviewer.

line 466: You never describe which analyses include which taxa and whether or not they are run with ordered or unordered characters. You need to be specific as to what was done in each analysis.

We have made this clear now.

line 548: This should be updated to the recent JHE article, not an abstract.

Pugh, K.D., 2022. Phylogenetic analysis of Middle-Late Miocene apes. *J. Hum. Evol.* 165, 103140

Done

Extended data annotated pdf:

line 27: How are these being defined? Maximum orbital breadth wherever it occurs or are there landmarks? Same question for all of the highlighted measurements. Postorbital breadth measure how? = Minimum frontal breadth? Or? Please define these somewhere in more specific terms or provide a reference that defined these measurements somewhere. Glabella-prosthion chord is pretty straightforward, but it is not clear how palatal breadth at M2, for example was measured. Is this internally or across the palate externally (i.e., biectomolare)? Etc. etc. You need an extra column with definitions for each measurement. Some of the measurements are defined in the Methods at the end of the main text and in various other captions and tables, but it would be better to have them all defined in one place.

Column added

line 35: What are the means of each group? This needs a little context here....I know that *Ouranopithecus* canines are small, and you have demonstrated significantly so. But please provide a table with the means (and ranges) for each group as well. Significance in the table, by itself, does not indicate the direction of the significant difference. But the means would easily make this clear.

Added.

line 42: You are listing the raw values in the table, but the PCA was run on each of these values divided by the GM, correct? Why not present the table using the actual corrected values plus the GM rather than the raw values?

Added

line 46: calculated from all 12 raw measurements included here, correct?

Clarified.

line 72: Ok, great. Which direction is which? Provide a bit more detail. e.g., something like "PC 1 is driven primarily by mandibular arch breadth such that broader mandibles are at the positive/negative end of PC1 and narrower mandibles are at the positive/negative end of PC1; PC2 is driven largely by symphyseal M1-M2 distance such that a longer symphysis is at the positive/negative end of PC2 and a shorter symphysis is at the negative/positive end of PC2."

Done

canine/gm plot: This is a more convincing plot than the canine/m2 one. The amount of variation in extant great apes in the first plot makes it clear that *Anadoluvius* could easily fit within a range of variation for *Ouranopithecus* in this metric and be in line with intrageneric (or intraspecific) variation. The canine/GM plot makes this much less likely.

We choose to include both plots because canine size is almost always compared with molars. We have added some text along the lines expressed by the reviewer. Thanks.

Again, canine area over GM? What is "canine"? Is m2 "size" m2 area (length x width)? The labels should be more specific as to what the metric is.

Clarified.

canine area / m2 area? Be specific as to what this metric is.

Specified.

So is this canine area / m2 area?

Clarified.

Again, size = area? Be specific here about what your metric is for size. In the first sentence, it reads as if only m2 is being expressed as an area.

Clarified.

line 101: Where? I don't see *Nakalipithecus* in Figure 4f. Please double check. And it should be corpus breadth / m2 size (as the square root of m2 area to make sure you are in linear units).

Our mistake. We included an earlier version of this figure. The revision includes the plot with *Nakalipithecus*.

line 150: Why map synapomorphies only for the unordered analysis? The ordered analysis needs to be provided as well. Also, provide some context here....this is the analysis from the main text, correct?

We have added mapping and Bremer supports for the ordered analysis as well. In the main text it is specified that this figure relates to the two 18 OTU analyses in the main text.

extended data figure 6a: *Graecopithecus* is not a hominine here, and a clade including *Graecopithecus*, *Ouranopithecus*, and *Anadoluvius* is not recovered. Also, provide more info on the analysis....how many taxa are analyzed? This is the expanded analysis relative to the one in the main text, correct? Is this ordered or unordered?

This cladogram is replaced with the reworking of the data matrix as per the reviewer's request. More details are provided on the details of the analysis of all 23 taxa in the text.

line 160: Again, more information in the caption please? How many taxa? Ordered or unordered? How does this relate to the main text?

Analysis completely revised

line 164: Same comment as above....provide more more information in the caption please

See revision

Supplemental materials annotated pdf:

line 138: This specimen should be included in the palatal comparison in Figure S5

That figure compares male specimens.

What figure number is this? Shouldn't this be Figure S1? If so, then all of the figure numberings need to be adjusted. Or maybe this could be integrated with current Fig. S3?

fixed

line 199: Yes, but they look much broader relative to their length than seen in CO 205. Can you describe the dentition here in more detail?

Detail added.

line 214: Are these both females? Please state the sex of each specimen so we know we are comparing apples to apples in terms of sexual dimorphism.

Sex identified in the revised legend.

line 244: Are these all supposed to be males? If so, please state that. Otherwise, I think the female palate from the cranium 2100/2800 should be included in this comparison.

All males (specified in the legend).

line 267: Are all these features still referring to I1? It's not clear as written.

Yes. Clarified.

line 297: and *Gigantopithecus*

Gigantopithecus is a much younger Pleistocene ape. It is obvious that the differences with Indopithecus also apply to Gigantopithecus, but we do not see the relevance in this context.

Table S4, character 7: Shouldn't this be 2 chars.? 1) Frontal sinus- States: 0=absent, 1=present and 2) Position of frontal sinus- 0=above glabella, 2=below glabella

We prefer to leave this as a single character. Absence of the frontal sinus is difficult to assess reliably given preservation issues.

All the highlighted characters are suggested by the reviewer to be ordered. We chose to order 21 or the 41, for reasons we explain in the methods. Basically, the characters that remain unordered have the intermediate state as the primitive condition.

Table S5: What is meant by these states? If being measured as minimum frontal breadth or something like that, then I would think states would be 0=great constriction (small relative min. frontal breadth), 1=moderate, 2=little constriction (large relative min. frontal breadth). Clarify what "deep" vs. wide means. And this character should be ordered if the sequence is great constriction to little constriction. The brain does not jump from very small to large without being intermediate in size first.

We have added more detailed definitions to all these characters and ordered many of them (see reply to review).

Character 8 should be 2 chars. First, whether a sinus is present or absent. Then, the position of the sinus when present.

We prefer to leave this as a single character. We would not be able to code the second character (position) in taxa lacking the sinus.

Table S5: When you ordered this, how was it ordered? I hope 0=narrow, 1=intermediate, 2=broad. In any case, the character states should be the same in the ordered and unordered analysis and should be listed here in an ordered fashion, even if left unordered. If the states change between the ordered and unordered analyses, then you would have to list the character states separately for that analysis, but I don't think your matrix can possibly line up to these states for both the unordered and ordered analyses as presented here.

We recoded all the characters from small to big, etc, that is, in an intuitive series from one extreme to the other with intermediates.

incisive canal: should probably be 2 chars: First incisive canal 0=absent, 1=present. Second is length of incisive canal when present 0=short, 2=long, 3=very long.

If kept as 1 char: 0=absent, 1=short, 2=long, 3=very long

We chose option two.

character 43: should be 2 characters; Foramen presence/absence 0=absent, 1=present; Foramen size when present 0=very small, 1=small, 2=large

See above. All characters recoded from one extreme to the other with intermediates.

Provide the nexus file as a separate SOM file for both the unordered and ordered matrices

Not sure what the reviewer wants here. This text is easily copied and pasted into TNT or any other software that accepts Nexus files. The ordering is done within the software. In TNT for example, one chooses which characters to order (additive) and which to leave unordered (non-additive) in character settings.

line 359: Are there other upper premolars and molars of Graecopithecus to compare it to? How would we know?

The reasons for attributing the Azmaka specimen to Graecopithecus are provided in the cited references.

Reviewers' comments:

Reviewer #2 (Remarks to the Author):

It is nice to see this paper continue to improve. Most of my suggestions were taken into account, so that's great. I have only a few relatively minor comments and some important clarifications that are needed. In addition, as you will see from my comments in the attached marked up MS Word documents, I continue to see numerous flaws in the logic behind the cladistic methods being presented. Because the authors express great exasperation and offense at some of my comments surrounding these methods in their "rebuttal" document, let me take a minute to clarify a few things upfront...

As I said in my last review, we are never going to see eye-to-eye on the cladistic analysis, and that is fine. The authors seem to take great personal offense at legitimate criticisms regarding the logic and reasoning behind their methodology, and that is unfortunate because the simple fact is this: if I am unconvinced using the modest powers of my little brain, I guarantee you that others will be unconvinced as well. My criticisms are meant to be constructive; if I have these concerns, one would think that the authors would like to try and address them or account for them as best they can because if I can find obvious logical flaws, then I guarantee others will find them as well. Instead, the authors seem deeply offended that I would dare to challenge their logic, and take things very personally, which is not very productive. And yes, I am passionate about getting things right and doing the best science we can, so I am passionate sometimes in my reviews when I feel that folks are not doing the best science they can and I think that illustrative examples help to convey my points as concretely as possible. I am not attacking anyone personally, but rather commenting on methods and logical ideas that seem to be very flawed as I read through the MS. But to be perfectly clear, I never recommended this paper to be rejected, so claiming that I am preventing publication due to philosophical differences is a completely unfounded accusation. I have always felt the paper should be published given the new genus being described and analyzed here, and I have said so to the editors. I have also been insistent that sections of the paper needed to be revised and/or better justified, as I continue to note here. I think the remaining revisions are relatively minor compared to before, and I expect this paper will be published in due course. But the continued resistance and borderline indignance to legitimate criticisms (not just philosophical differences) is unwarranted. I have given numerous options to the authors to satisfy my concerns and instead the authors continue to choose to plow ahead with an approach that is, in my opinion, full of serious logical flaws. For example, I suggested in my first review that the cladistic analysis simply be removed, but the authors did not heed this suggestion, which is, of course, their right to do. Instead, the authors presented a revised analysis that was still full of serious problems. But my objection has always been with the faulty logic and lack of details being presented...if the authors were to present a compelling argument for why their methodology is better or logically sound, then I would have no problem with it. As mentioned above, I even gave the option of removing the cladistic analysis here and citing previous studies instead. Or I provided a detailed list of all the characters that, in my view, should be ordered. If the authors chose to follow that advice and present both the unordered and recommended ordered analysis, I would also have no problem with the revision. But the fact is that these suggestions were largely side-stepped, and the authors have chosen to continue to make what are, in my view, poor arguments about why they should not order many of their multistate characters that would appear to make an obvious character transformation series. The current analysis is improved, to be sure, but it still has many of these same logical flaws. Again, the authors are free to make their own decisions, but my job as a reviewer is to offer my assessment to the editor and the authors as to whether the paper should be published and/or if it can or should be improved to make it as strong as possible. I have done my job to the best of my ability and in my view, the authors should again try to take a step back and understand that my comments are aimed at pointing out flaws in their logic and methodology that they should WANT to correct, not rejecting their paper due to philosophical differences.

On to the current paper. Again, a number of slight edits and more detailed comments are provided in the MS Word documents. My main concerns are summarized below:

- One of the biggest things that needs to be clarified is how many versions of the cladistic analysis were run. Throughout the main text, there are various references to an "All ordered" analysis and a "partly ordered" analysis. At first I thought there were 3 analyses: 1 unordered, 1 with 21 chars. ordered, and one "all ordered". However, the Methods describe only unordered and partially ordered iterations, so it looks like only two analyses were conducted: 1 unordered and one "partly ordered". Please be clear about exactly what analyses were conducted...If there are three analyses, then these need to be explicitly described and stated. In addition, if there was an analysis that ordered ALL multistate characters, then a number of the characters in Table S3 will need to be clearly identified and redefined so that the transformation series makes sense. For example, Char. 19 Nasal bone length. If this was ordered at any point, it needs to be 0=short, 1=long, 2=strongly elongated. If it was ordered as it is currently presented in Table S3 (0=long, 1=strongly elongate, 2=short), that is a problem ("strongly elongate" is not intermediate between long and short). The methods and iterations of the analyses must be clear. I realize that this is almost certainly due, in part, to the numerous revisions, but it still needs to be clarified.
- In the main text, page 19, the Nakalipithecus specimen (presumably the partial mandible) is referred to as probably male. In the SI, however, you say that the Nakalipithecus specimen is probably female. Which is it? If it is female (as often assumed), then this section in the main text needs to be re-written because it would fall within the range of Ouranopithecus females.
- Main text, Page 23: The GM is described as being derived from mandibular measurements and three dental dimensions. So how many measurements is that in total? I know you provide the measurements in the SI or Extended data, but please be specific here as well.
- Main text, Pages 25-26: At the risk of beating a dead horse, I find that the argument against ordering characters is still incredibly weak. Just because intermediate states aren't preserved in the fossil record due to sampling doesn't mean that they didn't exist. You seem to be modeling evolution based on the absence of fossil data. What happens when you find the fossil with intermediate maxillary length? Will you order it then? And if you would be willing to order it then, why not order it now if you know that it must have existed at some point? And what difference does it make if the outgroup has the "Intermediate" character state? It makes no difference as far as I can tell...there are many situations in which the primitive condition might very well be intermediate or more polymorphic between 2 fixed extremes within a given clade. But that doesn't change the fact that to move from one extreme to the other you would still have to move back through the intermediate morphological condition, and it only makes sense to model such characters in this way. Again, the reasoning here for why only 21 chars. were ordered makes very little sense.

Even the enamel thickness example is a weak argument. Yes, it is possible that a mutation might result in an individual having anomalously "thick" enamel relative to the rest of the population in a single generation. The issue is that evolution happens at the population level, not on the level of the individual. So this one thick-enamelled individual, if that mutation was advantageous, could pass on more copies of that allele (or alleles) to the next generation, and after many generations the result would be that the entire population would have intermediate enamel on average (the individuals with the thin alleles and now the individuals with the thick alleles being averaged together)! After enough generations, if thick enamel was still advantageous, the population would eventually be composed of only thick individuals, but there would have to be a period of time when the population was intermediate in enamel thickness. If you are imagining enamel thickness as discrete and Mendelian, then this example would be the result of a bimodal distribution, but if it is a codominant situation or the result of polygenic interactions, the more likely result is that the normal distribution curve for enamel thickness shifts towards thicker enamel gradually. In any case, an intermediate mean for the population would have to be passed through, so I just think this a poor argument all around. Again,

you have run analyses for a couple of different assumptions, which is what was greatly needed. You should be doing more, but have made it clear that you won't, which is unfortunate, but at least others can do so if they wish. But the reasoning for preferring the unordered analysis remains very weak. You are free to present it, but I don't think many will find this a very strong or convincing argument. I would simply remove it or find another way to justify your character modeling decisions.

- Main text, Page 27, vertebral count example: Again, see above. What you are suggesting in terms of "jumps" are not really possible at a population level. At some point, the population was variable and the average vertebral count for the population would shift through an intermediate value.

- Main text, Page 27, "Hypothesizing character state transformations that do not pass through intermediate states does not violate the basic principles of Darwinian evolution". For some characters, sure this is true...and I would agree that in those cases, unordering makes the most sense. But for any character for which there is a clear transformation series, then it would seem to me that it does violate basic evolutionary theory dating back to the Synthesis...Darwinian gradualism hypothesizes intermediates. They can evolve quickly or slowly, but the population is going to move in one direction or the other over time by passing through intermediates for any given trait, whether that be discrete traits or continuous traits. In the basic discrete Mendelian case, there will have to be a time with mixed alleles in the population, resulting in 2 different phenotypes...an intermediate "variable"/polymorphic population before fixing at one end or the other. In the continuous case, the population average will shift over time, passing through intermediate values. There may be a case to be made that there are other possibilities, but you have not laid out any convincing scenarios or evidence here.

- Extended data Table 4a: Interesting that the canine dimorphism of Ouranopithecus appears greater than Pan, even though the canines are relatively smaller in Ouranopithecus (i.e., male canine size/female canine size results in a larger value for Ouranopithecus compared to Pan). Doesn't this point to a unique type of canine reduction in Ouranopithecus compared to other taxa with reduced canines?

- Extended data Figure 6: The caption says this is "all ordered". Did you order all characters or only 21...you should be clear here. If they are all ordered, then they are not listed in any reasonable way in Table S3 and I have serious questions about the transformations being assumed. You really have to be clear about what you did here. It is not clear as currently written.

- Supplementary Materials, Table S3: See my previous comments above on the Methods and Results of your analyses from the main text. You must be clear about what you actually did here. I very much appreciate the added detail to the character definitions. But did you run an "All ordered" analysis? I assume the 21 ordered characters labeled here were included in the "partly ordered" analysis, but if you ordered more than this, please explain which ones and please explain or rewrite the character states in multistate characters to reflect a reasonable transformation series. For example, Char. 19 Nasal bone length. If this was ordered at any point, it needs to be 0=short, 1=long, 2=strongly elongated. If you ordered it as it is currently written (0=long, 1=strongly elongate, 2=short), that is a problem. You must clarify.

- Supplementary Materials, Table S3: Numerous of your characters that are labeled as "ordered" present transformation series that make no sense. For example, Character 7 (Supratoral sulcus). How is "pronounced" an intermediate morphology compared to 0=absent and 2=moderate? Shouldn't this be 0=absent, 1-moderate, 2=pronounced?

Similar examples are as follows:

Character 42 (Incisive foramen position): Shouldn't canine/P3 be intermediate between canine=0 and P3=2?

Character 51 (I1 cingulum): Again, the series described here doesn't make sense. Shouldn't this be 0=sharp, 1=mild, 2=absent? Why is absent currently listed as being intermediate?

Character 94 (p3 mesiobuccal flare): Shouldn't "intermediate" be the intermediate state between 0=strong and 2=weak here? Clarify.

If the analysis was run with the transformation series listed above, that is a problem. These characters and transformation series need to be clarified.

- Footnote to Supplementary Table S3:

The Pugh paper you cite here tested for and accounted for allometric effects while coding. Most characters were not influenced by allometry. Therefore, much of this comment is misleading and should be deleted. More broadly, I would think that quantitative data and more objectivity in character coding is a net positive, and that point is being ignored in the comment here. I don't see how you can argue that it is better to code everything qualitatively. This general perception has been abandoned at least since the Wiens (2000) volume on the phylogenetic analysis of morphological data. Even if coding quantitative data can be challenging, at least it is repeatable and different methods can always be employed. And different methods have been empirically demonstrated to generally produce more accurate trees (again see Wiens 2000 and other Wiens papers).

More broadly, I honestly don't see the point of trying to challenge a rival phylogenetic analysis in a footnote. If you want to challenge the Pugh analysis, then be upfront about it and discuss it in the main text. I don't think many will find arguments against coding with quantitative data convincing, but you are free to give it a shot. In any case, it should not be done in a footnote.

One of the biggest things that needs to be clarified is how many versions of the cladistic analysis were run. Throughout the main text, there are various references to an “All ordered” analysis and a “partly ordered” analysis. At first I thought there were 3 analyses: 1 unordered, 1 with 21 chars. ordered, and one “all ordered”. However, the Methods describe only unordered and partially ordered iterations, so it looks like only two analyses were conducted: 1 unordered and one “partly ordered”. Please be clear about exactly what analyses were conducted...If there are three analyses, then these need to be explicitly described and stated. In addition, if there was an analysis that ordered ALL multistate characters, then a number of the characters in Table S3 will need to be clearly identified and redefined so that the transformation series makes sense. For example, Char. 19 Nasal bone length. If this was ordered at any point, it needs to be 0=short, 1=long, 2=strongly elongated. If it was ordered as it is currently presented in Table S3 (0=long, 1=strongly elongate, 2=short), that is a problem

(“strongly elongate” is not intermediate between long and short). The methods and iterations of the analyses must be clear. I realize that this is almost certainly due, in part, to the numerous revisions, but it still needs to be clarified.

We go into some detail explaining why we did not order characters in which the primitive (outgroup) condition is the intermediate one. Ordering would force the algorithm to jump from one extreme to the other.

- In the main text, page 19, the *Nakalipithecus* specimen (presumably the partial mandible) is referred to as probably male. In the SI, however, you say that the *Nakalipithecus* specimen is probably female. Which is it? If it is female (as often assumed), then this section in the main text needs to be re-written because it would fall within the range of *Ouranopithecus* females.

A more careful reading of our text shows clearly that we are referring to different specimens of *Nakalipithecus* in the main text and in the SI (the molar from the large, almost certainly male mandible vs other much smaller and probably female specimens not associated with the mandible. The identification of the specimens are clear in the text.

- Main text, Page 23: The GM is described as being derived from mandibular measurements and three dental dimensions. So how many measurements is that in total? I know you provide the measurements in the SI or Extended data, but please be specific here as well.

We believe we have adequately defined the GM throughout.

- Main text, Pages 25-26: At the risk of beating a dead horse, I find that the argument against ordering characters is still incredibly weak. Just because intermediate states aren't preserved in the fossil record due to sampling doesn't mean that they didn't exist. You seem to be modeling evolution based on the absence of fossil data. What happens when you find the fossil with intermediate maxillary length? Will you order it then? And if you would be willing to order it then, why not order it now if you know that it must have existed at some point? And what difference does it make if the outgroup has the “Intermediate” character state? It makes no difference as far as I can tell...there are many situations in which the primitive condition might very well be intermediate or more polymorphic between 2 fixed extremes within a given clade. But that doesn't change the fact that to move from one extreme to the

other you would still have to move back through the intermediate morphological condition, and it only makes sense to model such characters in this way. Again, the reasoning here for why only 21 chars. were ordered makes very little sense.

There is no further response to this dogmatic position about the absolute necessity that all transformations must pass through intermediate states beyond the arguments we have already presented. It is naïve to assume that we know enough about the genetics controlling morphology to say that all transformations must pass through a given trajectory. In the end, extremes and intermediates are artificial constructs of our apprehension of morphology. They are not based on a complete and documented theory of the actual genetic basis of character transformation. Because we see a character state as intermediate does not mean that from a genetic perspective the only transformation series possible is through that particular character state.

Even the enamel thickness example is a weak argument. Yes, it is possible that a mutation might result in an individual having anomalously “thick” enamel relative to the rest of the population in a single generation. The issue is that evolution happens at the population level, not on the level of the individual. So this one thick-enamelled individual, if that mutation was advantageous, could pass on more copies of that allele (or alleles) to the next generation, and after many generations the result would be that the entire population would have intermediate enamel on average (the individuals with the thin alleles and now the individuals with the thick alleles being averaged together)! After enough generations, if thick enamel was still advantageous, the population would eventually be composed of only thick individuals, but there would have to be a period of time when the population was intermediate in enamel thickness. If you are imagining enamel thickness as discrete and Mendelian, then this example would be the result of a bimodal distribution, but if it is a codominant situation or the result of polygenic interactions, the more likely result is that the normal distribution curve for enamel thickness shifts towards thicker enamel gradually. In any case, an intermediate mean for the population would have to be passed through, so I just think this a poor argument all around. Again, you have run analyses for a couple of different assumptions, which is what was greatly needed. You should be doing more, but have made it clear that you won’t, which is unfortunate, but at least others can do so if they wish. But the reasoning for preferring the unordered analysis remains very weak. You are free to present it, but I don’t think many will find this a very strong or convincing argument. I would simply remove it or find another way to justify your character modeling decisions.

We responded to this rather rigid perspective of gradualistic evolution and will have to agree to disagree.

- Main text, Page 27, vertebral count example: Again, see above. What you are suggesting in terms of “jumps” are not really possible at a population level. At some point, the population was variable and the average vertebral count for the population would shift through an intermediate value.
- Main text, Page 27, “Hypothesizing character state transformations that do not pass through intermediate states does not violate the basic principles of Darwinian evolution”. For some characters, sure this is true...and I would agree that in those cases, unordering makes the most sense. But for any character for which there is a clear transformation series, then it would seem to me that it does violate basic evolutionary theory dating back to the Synthesis....Darwinian gradualism hypothesizes intermediates. They can evolve quickly or slowly, but the population is going to move in one direction or the other over time by passing through intermediates for any given trait, whether that be discrete traits

or continuous traits. In the basic discrete Mendelian case, there will have to be a time with mixed alleles in the population, resulting in 2 different phenotypes...an intermediate “variable”/polymorphic population before fixing at one end or the other. In the continuous case, the population average will shift over time, passing through intermediate values. There may be a case to be made that there are other possibilities, but you have not laid out any convincing scenarios or evidence here.

See above

- Extended data Table 4a: Interesting that the canine dimorphism of *Ouranopithecus* appears greater than *Pan*, even though the canines are relatively smaller in *Ouranopithecus* (i.e., male canine size/female canine size results in a larger value for *Ouranopithecus* compared to *Pan*). Doesn't this point to a unique type of canine reduction in *Ouranopithecus* compared to other taxa with reduced canines?

Or a unique type of canine reduction in *Pan* compared with *Ouranopithecus* and hominines. The argument could go both ways. However, given the small samples sizes we are not comfortable quantifying the degree of sexual dimorphism in *Ouranopithecus* canines. In addition, the metrics refer to dimensions are the crown cervix and not canine height, so they incompletely represent overall canine dimorphism (which was not the goal of that analysis, which is focused on relative canine size regardless of sex). Therefore, we consider it more advisable not to prematurely address this point at this time.

- Extended data Figure 6: The caption says this is “all ordered”. Did you order all characters or only 21...you should be clear here. If they are all ordered, then they are not listed in any reasonable way in Table S3 and I have serious questions about the transformations being assumed. You really have to be clear about what you did here. It is not clear as currently written.

We have added “fully” to the legend.

- Supplementary Materials, Table S3: See my previous comments above on the Methods and Results of your analyses from the main text. You must be clear about what you actually did here. I very much appreciate the added detail to the character definitions. But did you run an “All ordered” analysis? I assume the 21 ordered characters labeled here were included in the “partly ordered” analysis, but if you ordered more than this, please explain which ones and please explain or rewrite the character states in multistate characters to reflect a reasonable transformation series. For example, Char. 19 Nasal bone length. If this was ordered at any point, it needs to be 0=short, 1=long, 2=strongly elongated. If you ordered it as it is currently written (0=long, 1=strongly elongate, 2=short), that is a problem. You must clarify.

We hope we have clarified this in the methods.

Supplementary Materials, Table S3: Numerous of your characters that are labeled as “ordered” present transformation series that make no sense. For example, Character 7 (Supratoral sulcus). How is “pronounced” an intermediate morphology compared to 0=absent and 2=moderate? Shouldn't this be 0=absent, 1-moderate, 2=pronounced?

Similar examples are as follows:

Character 42 (Incisive foramen position): Shouldn't canine/P3 be intermediate between canine=0 and P3=2?

Character 51 (I1 cingulum): Again, the series described here doesn't make sense. Shouldn't this be 0=sharp, 1=mild, 2=absent? Why is absent currently listed as being intermediate?

Character 94 (p3 mesiobuccal flare): Shouldn't "intermediate" be the intermediate state between 0=strong and 2=weak here? Clarify.

If the analysis was run with the transformation series listed above, that is a problem. These characters and transformation series need to be clarified.

I am not sure "numerous" is the right word for four instances, but in any case, these are typos. We changed these in the data matrix but missed them in the character descriptions. These have been corrected. We thank the reviewer for their careful reading.

Footnote to Supplementary Table S3:

The Pugh paper you cite here tested for and accounted for allometric effects while coding. Most characters were not influenced by allometry. Therefore, much of this comment is misleading and should be deleted. More broadly, I would think that quantitative data and more objectivity in character coding is a net positive, and that point is being ignored in the comment here. I don't see how you can argue that it is better to code everything qualitatively. This general perception has been abandoned at least since the Wiens (2000) volume on the phylogenetic analysis of morphological data. Even if coding quantitative data can be challenging, at least it is repeatable and different methods can always be employed. And different methods have been empirically demonstrated to generally produce more accurate trees (again see Wiens 2000 and other Wiens papers).

More broadly, I honestly don't see the point of trying to challenge a rival phylogenetic analysis in a footnote. If you want to challenge the Pugh analysis, then be upfront about it and discuss it in the main text. I don't think many will find arguments against coding with quantitative data convincing, but you are free to give it a shot. In any case, it should not be done in a footnote.

Our goal was not to challenge "a rival phylogenetic analysis", which incidentally is not how we see the excellent contribution by Pugh (2022). It is an alternative hypothesis, as we note. Our intention was to point to the important finding in Pugh (2022) that discretized characters based on measurements are problematic and to note that simple ratios fail to account for allometry. There is no mention of testing for allometric effects in Pugh (2022). In fact, the words allometry and allometric do not appear in the text of Pugh (2022). We did make a modification to this footnote, adding that Pugh uses both "related structures" and first molar size to generate ratios.